# Abstracted Shapes as Tokens - A Generalizable and Interpretable Model for Time-series Classification

**Yunshi Wen** *
Rensselaer Polytechnic
Institute
weny2@rpi.edu

**Tengfei Ma**\*
Stony Brook University
tengfei.ma@stonybrook.edu

**Tsui-Wei Weng**
University of California,
San Diego
lweng@ucsd.edu

**Lam M. Nguyen**
IBM Research
lamnguyen.mltd@ibm.com

**Anak Agung Julius**
Rensselaer Polytechnic
Institute
agung@ecse.rpi.edu

## Abstract

In time-series analysis, many recent works seek to provide a unified view and representation for time-series across multiple domains, leading to the development of foundation models for time-series data. Despite diverse modeling techniques, existing models are black boxes and fail to provide insights and explanations about their representations. In this paper, we present VQShape, a pre-trained, generalizable, and interpretable model for time-series representation learning and classification. By introducing a novel representation for time-series data, we forge a connection between the latent space of VQShape and shape-level features. Using vector quantization, we show that time-series from different domains can be described using a unified set of low-dimensional codes, where each code can be represented as an abstracted shape in the time domain. On classification tasks, we show that the representations of VQShape can be utilized to build interpretable classifiers, achieving comparable performance to specialist models. Additionally, in zero-shot learning, VQShape and its codebook can generalize to previously unseen datasets and domains that are not included in the pre-training process. The code and pre-trained weights are available at `https://github.com/YunshiWen/VQShape`.

## 1 Introduction

As one of the fundamental forms of data, time-series (TS) exist in a wide range of domains and applications, including healthcare, weather, traffic, motions, human activities, sensors, etc. Modeling TS data across multiple domains has been a challenging task since TS data can have diverse sampling rates, lengths, magnitudes, frequencies, and noise levels. Due to this heterogeneity, most of the existing machine learning methods for TS modeling focus only on a single dataset or a single domain.

Recently, motivated by the success of large pre-trained models in natural language processing and computer vision, various approaches adopted from these two fields have been proposed to build a unified view and feature space for TS data from different domains [Liang et al., 2024]. Most of the models use a transformer as the backbone and pre-train it on a diverse range of datasets [Zerveas et al., 2021, Nie et al., 2023, Goswami et al., 2024]. These methods have achieved great success in TS representation learning, benefiting various downstream tasks and demonstrating their

---

*Corresponding to Yunshi Wen (weny2@rpi.edu) and Tengfei Ma (tengfei.ma@stonybrook.edu)

38th Conference on Neural Information Processing Systems (NeurIPS 2024).

generalizability. Despite their success, most of them remain black boxes since they cannot provide human-understandable representations. While tokenizers have played increasingly important roles in pre-trained models for language and vision, in TS, pre-training is often conducted by predicting the next or masked timestamp, time window, or patch, lacking the concept of discrete tokens as in LLMs. Very recently, Talukder et al. [2024] developed TOTEM, which utilizes VQ-VAE [van den Oord et al., 2017] to obtain the codebook and reconstruct the TS. Nevertheless, like all other VQ-VAE models, the tokens from the codebook are just latent vector representations and lack physical meaning.

Alternatively, in interpretable TS modeling, shapelets have been recognized as interpretable and expressive features for TS data. Initially defined as TS subsequences that discriminate different categories in classification [Ye and Keogh, 2011], they were later generalized to representative patterns [Grabocka et al., 2014]. Specifically, shapelets can transform TS data into low-dimensional representations either in the form of the distance between a shapelet and a TS, or as a logical predicate that measures the probability of a shapelet existing in a TS [Lines et al., 2012]. However, despite their effectiveness in classification tasks, this shape-level feature lacks flexibility since shapelets with pre-defined lengths are optimized for capturing discriminative features for making dataset-specific predictions. For example, when measuring human motion with accelerometers, an adult and a child performing the same gesture may record TS with different offsets, scales, and durations. Although they share the same shape-level concept, multiple shapelets are required to describe them separately. Additionally, shapelet-based interpretable models are specialized to a single dataset, and the learned shapelets fail to transfer to different domains.

In this paper, motivated by the limitations of existing pre-trained models and interpretable models in TS, we propose VQShape, a self-supervised pre-trained model that provides abstracted shapes as interpretable and generalizable tokens for TS modeling. Firstly, we decompose a TS subsequence into a set of attributes, including abstracted shape, offset, scale, start time, and duration. By incorporating vector quantization, VQShape learns a codebook of abstracted shapes that are generalizable and descriptive, representing TS from various domains. Evaluated on various classification tasks, and without fine-tuning, VQShape achieves comparable performance to black-box pre-trained models while additionally providing interpretable latent-space tokens and representations to describe TS data. Our contributions are summarized below:

- We present an interpretable representation composed of abstracted shapes and attributes to describe TS data based on shape-level features, which enables the learning of dataset-agnostic interpretable features.

- We introduce VQShape, to the best of our knowledge, the first self-supervised pre-trained model that extracts interpretable representations from any TS data. VQShape also learns a codebook containing abstracted shapes that generalize to multiple datasets.

- Pre-trained on diverse datasets and without fine-tuning, VQShape achieves comparable performance to existing black-box models on benchmark classification datasets. We explicitly demonstrate that the representations and VQShape are interpretable and generalizable for unseen datasets and domains.

## 2 Related Work

**Deep learning methods for TS analysis.** Deep learning methods are increasingly applied to TS analysis. Existing methods can be categorized into two groups depending on whether they use a Transformer structure as the backbone. For non-Transformer-based models, classical deep learning models such as MLP, CNN, and ResNet demonstrate decent performance on various tasks [Wang et al., 2017]. Recent methods have developed various feature engineering techniques to model explicit features of TS data. TimesNet [Wu et al., 2023] transforms TS into 2D space to capture multi-period features in a modularized way, achieving state-of-the-art performance on various tasks. TS2Vec [Yue et al., 2022] employs hierarchical contrastive learning for unsupervised representation learning of TS data. T-Rep [Fraikin et al., 2024] introduces a self-supervised representation learning approach by augmenting the TS with time embeddings, providing additional temporal structure to the latent space.

Transformers have been increasingly applied to TS analysis, but usually with some modifications to the original structure. For example, Autoformer [Wu et al., 2021] modifies the attention mechanism by incorporating an Auto-Correlation mechanism to capture temporal dependencies. When applying Transformers to real-valued data, transforming the inputs into patches has been recognized as an

effective approach for images [Dosovitskiy et al., 2021] since the tokens could contain more semantic meaning, like a "word" in language. Similarly, PatchTST [Nie et al., 2023] shows that TS analysis also benefits from combining patched inputs with Transformers, viewing a TS as a sequence of 64 "words".

**Pre-trained Models for TS data.** The success of large pre-trained models in language and vision motivates the development of foundation models for TS analysis. Existing approaches aim to find a unified view for TS data from different perspectives. For example, TST [Zerveas et al., 2021] uses the Transformer model [Vaswani et al., 2017] and is pre-trained using masked reconstruction, while TimeGPT-1 [Garza et al., 2023] is pre-trained by generating a forecasting window. MOMENT [Goswami et al., 2024] extends a patch-based Transformer [Nie et al., 2023] to multiple datasets by unifying the lengths of TS data using padding and sub-sampling. The model is also pre-trained to reconstruct the masked patches. TOTEM [Talukder et al., 2024] applies a convolutional neural network (CNN) encoder to raw TS data and uses vector quantization (VQ) on the encoder outputs, providing a discrete and domain-invariant codebook for TS data. TOTEM is pre-trained as a VQ-VAE [van den Oord et al., 2017] to reconstruct the whole TS, viewing the latent-space codes from convolutions as a unified representation. UniTS [Gao et al., 2024] introduces a prompt-based method to unify predictive and generative tasks within a single model and pre-training process. Although these methods learn representations that benefit various downstream tasks and demonstrate generalizability, these pre-trained models remain black boxes since they cannot provide human-understandable representations.

## 3 Proposed Method

Towards interpretable TS modeling, we first present the formulations of shape-level representations, describing univariate TS data using a set of abstracted shapes and attributes. Then, we introduce the architecture of VQShape and its components with detailed workflow and products from each step.

**Notations.** Let $(\mathcal{X}, \mathcal{Y}) = \{(x_i, y_i) | i = 1, \ldots, N\}$ denote a TS classification dataset with $N$ samples, where $x_i \in \mathbb{R}^{M \times T}$ is a multivariate TS sample and $y_i \in \{1, \ldots, C\}$ is the class label. Here, $M$ is the number of variables, $T$ is the length in timestamp, and $C$ is the number of categories. Each multivariate TS sample $x_i$ can be viewed as a set of univariate TS samples where $x_i^m \in \mathbb{R}^T$ denotes the TS at the $m^{\text{th}}$ variable. For simplicity in notations, in this paper, $x_{i,t_1:t_2}^m$ denotes a subsequence of $x_i^m$ between timestamp $\lfloor T t_1 \rfloor$ and $\lfloor T t_2 \rfloor$, where $t_1, t_2 \in [0, 1]$ are relative positions.

### 3.1 Shape-level representation

For a univariate TS $x$, a subsequence $s_k$ can be represented by an attribute tuple $\tau_k = (z_k, \mu_k, \sigma_k, t_k, l_k)$ where

- $z_k \in \mathbb{R}^{d_{\text{code}}}$ is the code for abstracted shape of $s_k$,
- $\mu_k \in \mathbb{R}^1$ is the offset of $s_k$,
- $\sigma_k \in \mathbb{R}^1$ is the scale (standard deviation) of $s_k$ and $\sigma_k > 0$,
- $t_k \in \mathbb{R}^1$ is the relative starting position of $s_k$ in $x$ and $0 \leq t_k \leq 1 - l_{\min}$,
- $l_k \in \mathbb{R}^1$ is the relative length of $s_k$ w.r.t. the length of $x$ and $l_{\min} \leq l_k \leq 1 - t_k$.

Here, $l_{\min}$ is the hyperparameter that defines the minimum length of a shape. We set $l_{\min} = 1/64$ as it is the length of a patch. In this work, we develop a pre-trained transformer model to produce a set of attribute tuples $\mathcal{T} = \{\tau_k \mid k = 1, \ldots, K\}$ given a univariate TS $x$. Additionally, the model learns a codebook of abstracted shape $z$ that is reusable and generalizable for datasets from different domains.

### 3.2 VQShape Architecture

The VQShape model contains a TS encoder $\mathcal{E}$, a TS decoder $\mathcal{D}$, a latent-space codebook $\mathcal{Z}$, a shape decoder $\mathcal{S}$, an attribute encoder $\mathcal{A}_{\text{enc}}$, and an attribute decoder $\mathcal{A}_{\text{dec}}$. An overview of VQShape is presented in Figure 1. We then present a detailed formulation for each component.

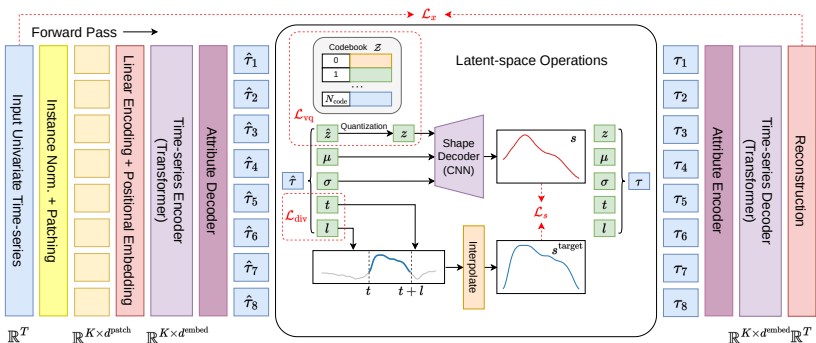

Figure 1: Overview of VQShape

**TS Encoding.** VQShape contains a patch-based transformer encoder [Nie et al., 2023, Goswami et al., 2024] which first transforms a univariate TS $x$ into $K$ non-overlapping fixed-length patches with dimension $d^{\text{patch}}$. Then, the patches are encoded by learnable linear projection and additive position embedding, forming patch embeddings that serve as inputs to a transformer model. The transformer outputs $K$ latent embeddings $\hat{h} \in \mathbb{R}^{d^{\text{embed}}}$. Formally, the TS encoder is denoted by $\{\hat{h}_k \in \mathbb{R}^{d^{\text{embed}}} \mid k = 1, \ldots, K\} = \mathcal{E}(x)$. Note that $\hat{h}_k$ could contain information from all patches instead of only the $k^{\text{th}}$ patch.

**Attribute Decoding.** The attribute decoder $\mathcal{A}_{\text{dec}}$ takes a latent embedding $h_k$ and extracts an attribute tuple $\hat{\tau}_k = (\hat{z}_k, \mu_k, \sigma_k, t_k, l_k)$. Formally, $\mathcal{A}_{\text{dec}}$ performs

$$\hat{\tau}_k = (\hat{z}_k, \mu_k, \sigma_k, t_k, l_k) = \mathcal{A}_{\text{dec}}(h_k), \text{ where } \begin{cases} \hat{z}_k = f_z(h_k), \\ \mu_k = f_\mu(h_k), \\ \sigma_k = \texttt{softplus}(f_\sigma(h_k)), \\ t_k = \texttt{sigmoid}(f_t(h_k)) \cdot (1 - l_{\min}), \\ l_k = \texttt{sigmoid}(f_l(h_k)) \cdot (1 - t_k) + l_{\min}. \end{cases} \quad (1)$$

Each decoding function in $\{f_z, f_\mu, f_\sigma, f_t, f_l\}$ is implemented using a multi-layer perceptron (MLP) with one hidden layer and $\texttt{ReLU}$ activation. Following a common notation [Esser et al., 2021], $\hat{\tau}$ denotes the attribute tuple before quantization.

**Codebook and Vector-Quantization.** The latent-space codebook is denoted by $\mathcal{Z} = \{z_q \in \mathbb{R}^{d^{\text{code}}} \mid q = 1, \ldots, N^{\text{code}}\}$. To learn a generalizable codebook that contains only the abstracted shape-level features, we use low-dimensional codes with $d^{\text{code}} = 8$. This configuration also creates a bottleneck for reconstruction, minimizing additional information that can be inferred besides the abstracted shapes. The quantization follows VQ-VAE [van den Oord et al., 2017] that selects the discrete code based on Euclidean distance where

$$z_k = \underset{z_q \in \mathcal{Z}}{\arg\min} \|\hat{z}_k - z_q\|. \quad (2)$$

**Shape Decoding.** The abstracted shape of a TS subsequence is a sequence with its length, offset, and scale information removed through normalizations. Given $\tau_k = (z_k, \mu_k, \sigma_k, t_k, l_k)$, we first extract the target subsequence from $x$ specified by $t_k$ and $l_k$ denoted by $x_{t_k:t_k+l_k}$. Then, $x_{t:t+l}$ is interpolated to a fixed length of $d^s$ to remove the length information. The shape decoder $\mathcal{S}$ takes $z_k$ and outputs another sequence with the same length. Formally, for $\tau_k$, this step produces two sequences

$$\begin{aligned} s_k^{\text{target}} \in \mathbb{R}^{d^s} &= \texttt{interpolate}(x_{t_k:t_k+l_k}), \\ s_k \in \mathbb{R}^{d^s} &= \mathcal{S}(z_k) \cdot \sigma_k + \mu_k. \end{aligned} \quad (3)$$

Note that the output of $\mathcal{S}$ is normalized such that $\mathcal{S}(z_k)$ has the offset and scale information removed.

**Attribute encoding and reconstruction.** The attribute tuple after quantization $\tau_k = (z_k, \mu_k, \sigma_k, t_k, l_k)$ is transformed by a learnable linear projection denoted by $h_k \in \mathbb{R}^{d^{\text{embed}}} = \texttt{Linear}(\tau_k)$. Then, the TS decoder $\mathcal{D}$ takes $\{h_k \mid k = 1, \ldots, K\}$ and outputs the reconstructed TS $\hat{x}$.

## 4 Pre-training

VQShape is pre-trained on diverse datasets to learn dataset-agnostic features and tokens. In this section, we introduce the self-supervised training strategies and objectives of VQShape. Then, we discuss the representations the model could provide to down-stream tasks.

### 4.1 Objectives

The optimization objectives of VQShape during the pre-training stage are summarized below.

**Reconstructions.** Analogous to most of the VQ-VAE approaches, VQShape is trained to accurately reconstruct the input TS to learn essential latent-space representations for modeling TS data. Additionally, to provide interpretable representations, the decoded shapes should be similar to the actual subsequences. Therefore, the reconstruction minimizes two objectives:

$$\text{Time-series reconstruction:} \qquad \mathcal{L}_x = \|x - \hat{x}\|_2^2, \tag{4}$$

$$\text{Subsequence reconstruction:} \qquad \mathcal{L}_s = \frac{1}{K} \sum_{k=1}^{K} \|s_k^{\text{target}} - s_k\|_2^2. \tag{5}$$

**Vector Quantization.** We follow VQ-VAE [van den Oord et al., 2017] to define the vector-quantization objective which trains the encoder $\mathcal{E}$ and codebook $\mathcal{Z}$. Additionally, inspired by Yu et al. [2024], we add additional entropy terms to encourage codebook usage. We find these terms could improve pre-training stability and avoid collapse of codebook usage. The objective for learning the codebook is defined by

$$\mathcal{L}_{\text{vq}} = \underbrace{\|\hat{z} - \texttt{sg}(z)\|_2^2 + \lambda_{\text{commit}}\|\texttt{sg}(\hat{z}) - z\|_2^2}_{\text{quantization}} + \underbrace{\mathbb{E}\left[H(q(\hat{z}, \mathcal{Z}))\right] - H(\mathbb{E}\left[q(\hat{z}, \mathcal{Z})\right])}_{\text{codebook usage}}, \tag{6}$$

where $\texttt{sg}(\cdot)$ is the stop-gradient operator and $H(\cdot)$ is the entropy function for discrete variables. $q(z, \mathcal{Z}) = \texttt{softmax}\left(\left[\|\hat{z} - z_k\|_2^2 \mid z_k \in \mathcal{Z}\right]\right)$ measures the distance between $\hat{z}$ and all codes in $\mathcal{Z}$ as a categorical distribution.

**Disentanglement of shapes.** In Equation 5, the attributes $(z_k, \mu_l, \sigma_k)$ are optimized towards accurate subsequence reconstructions. It is important to note that, since $(t_k, l_k)$ defines $s_k^{\text{target}}$, they are essential for learning the abstracted shapes and the codebook. However, it is challenging to use gradients from reconstruction in Equation 4 solely to learn $(t_k, l_k)$ for making informative subsequence selection. Therefore, we introduce an additional regularization that encourages the latent-space tokens (attributes) to capture shape-level information with diverse positions and scales. This regularization is defined as

$$\mathcal{L}_{\text{div}} = \frac{1}{K^2} \sum_{k_1=1}^{K} \sum_{k_2=1}^{K} \mathbb{1}(k_1 \neq k_2)\, \texttt{relu}\left(\epsilon - \|\kappa(t_{k_1}, l_{k_1}) - \kappa(t_{k_2}, l_{k_2})\|_2^2\right),$$

$$\text{where } \kappa(t_k, l_k) = \begin{bmatrix} \cos(t_k \pi) \cdot \ln(l_k)/\ln(l_{\min}) \\ \sin(t_k \pi) \cdot \ln(l_k)/\ln(l_{\min}) \end{bmatrix}. \tag{7}$$

In Equation 7, $\kappa(t_k, l_k)$ defines a coordinate transformation which maps $(t_k, l_k)$ into a space where (1) small $l_k$ values become more diverse and (2) large $l_k$ values from different $t_k$ become more concentrated. By making different $(t_k, l_k)$ diverse in this space, $\mathcal{L}_{\text{div}}$ encourages the model to capture disentangled shape-level information while increasing the use of short sequences to capture local details. Figure 8 visualizes an example of transformation $\kappa$. $\epsilon$ is a hyperparameter that defines a threshold distance in the transformed coordinate where two $(t_k, l_k)$ samples are considered sufficiently diverse.

The overall pre-training objective is to minimize

$$\mathcal{L}_{\text{pretrain}} = \lambda_x \mathcal{L}_x + \lambda_s \mathcal{L}_s + \lambda_{\text{vq}} \mathcal{L}_{\text{vq}} + \lambda_{\text{div}} \mathcal{L}_{\text{div}}, \tag{8}$$

where $\lambda_x, \lambda_s, \lambda_{\text{vq}}, \lambda_{\text{div}}$ are hyperparameters that define the weighting between the components. During pre-training of VQShape, we set $\lambda_x = \lambda_s = \lambda_{\text{vq}} = 1$, $\lambda_{\text{div}} = 0.8$, and $\lambda_{\text{commit}} = 0.25$.

**Design Analysis.** Overall, the encoding process in VQShape (Transformer encoder and attribute decoder) introduces an inductive bias by representing and summarizing univariate TS using a set of abstracted shapes along with their position, length, offset, and scale. The pre-training objectives guide the components toward learning interpretable representations (via subsequence reconstruction in Equation 5) and disentangled representations (via regularization in Equation 7), while preserving the information necessary to describe the TS (via reconstruction in Equation 4). These objectives introduce interpretability to the conventional deep autoencoder structure. By pre-training on diverse datasets with a universal codebook, VQShape further leverages this inductive bias to produce discrete and dataset-agnostic representations, resulting in a vocabulary of abstracted shapes that can be used as primitives to describe TS data.

**Model Configurations.** The settings of VQShape related to the model size correspond to those of the `MOMENT-Small` [Goswami et al., 2024] model. Specifically, we interpolate all the input univariate TS $x$ to have length $T = 512$, which is broken into $K = 64$ patches with $d^{\text{patch}} = 8$. The Transformer layers in the encoder $\mathcal{E}$ and decoder $\mathcal{D}$ have 8 heads, an embedding dimension $d^{\text{embed}} = 512$, and a feed-forward layer of size $2048$. We employ an asymmetric structure with an 8-layer encoder $\mathcal{E}$ and a 2-layer decoder $\mathcal{D}$ [He et al., 2022]. The codebook $\mathcal{Z}$ contains $N^{\text{code}} = 512$ codes, each with dimension $d^{\text{code}} = 8$. The subsequences $s_k^{\text{target}}$ and decoded sequences $s_k$ have length $d^s = 128$. We set the minimum shape length $l\text{min} = 1/64$. With these settings, VQShape has 37.1 million parameters.

In the pre-training stage, we train VQShape with the AdamW optimizer, using weight decay $\lambda = 0.01$, $\beta_1 = 0.9$, $\beta_2 = 0.999$, gradient clipping of $1.0$, and an effective batch size of $2048$. We employ a cosine learning rate schedule with an initial learning rate of $1e^{-4}$, a final learning rate of $1e^{-5}$, and 1 epoch of linear warm-up. The pre-training dataset contains univariate TS extracted from the training split of 29 datasets from the UEA Multivariate TS Classification Archive [Bagnall et al., 2018], excluding the `InsectWingbeat` dataset, resulting in 1,387,642 univariate TS. We train VQShape for 50 epochs on this dataset using `bfloat-16` mixed precision.

### 4.2 Representations for down-stream tasks

VQShape provides two types of representations: **Latent-space Tokens** and **Code Histogram**.

**Tokens.** Similar to the latent-space feature map of typical VQ approaches such as VQ-VAE [van den Oord et al., 2017] and VQ-GAN [Esser et al., 2021], VQShape also provides a set of tokens as representations. For an input univariate TS $x$, the token representations are composed as $\mathcal{T} \in \mathbb{R}^{K \times (d^{\text{code}}+4)} = \{\tau_k = (z_k, \mu_k, \sigma_k, t_k, l_k) \mid k = 1, \ldots, K\}$. The token representations can be useful for general down-stream tasks but are less interpretable than the code histogram representations in classification tasks.

**Code Histogram.** Inspired by Concept Bottleneck Models (CBMs) [Koh et al., 2020] developed in computer vision, we can also view each $z_q \in \mathcal{Z}$ as a concept for TS data. As CBMs have concept scores as representations, VQShape provides a similar representation in the form of a histogram of codes. Based on Equation 2, we can also have a vector of code indices

$$\mathbf{q} = \left[ q_k = \underset{q=1,\ldots,N^{\text{code}}}{\arg\min} \|\hat{z}_k - z_q\| \mid k = 1, \ldots, K \right]. \tag{9}$$

Then, the code histogram representation is defined as $\mathbf{r} \in \mathbb{R}^{N^{\text{code}}} = \texttt{histogram}(\mathbf{q})$ where each element in $\mathbf{r}$ is the frequency of index $q$ in $\mathbf{q}$. Intuitively, the code histogram representation is analogous to BOSS [Schäfer, 2015] but with non-deterministic window size and dataset-agnostic symbols. In classification tasks, this type of representation can be more interpretable since classifiers based on these features are able to produce rule-like predictions that are straightforward to interpret and understand.

# 5 Experiments

**Datasets.** We evaluate the pre-trained VQShape on multivariate TS classification tasks to demonstrate the effectiveness of learned shape-level representations on down-stream tasks. The evaluations are performed on the test split of 29 datasets from the UEA multivariate TS classification archive [Bagnall et al., 2018, Ruiz et al., 2021], with the `InsectWingbeat` dataset excluded. Table 4 summarizes statistics of those datasets. Details on experiment setups are included in Appendix A.

**Baselines.** We benchmark VQShape against baselines from four categories, including (1) **classical** methods: DTW [Chen et al., 2013] and Shapelet Transform with Random Forest classifier (STRF) [Bostrom and Bagnall, 2017], (2) **supervised** learning methods: DLinear [Zeng et al., 2023], Autoformer [Wu et al., 2021], FEDformer [Zhou et al., 2022], PatchTST [Nie et al., 2023], and TimesNet [Wu et al., 2023], (3) **unsupervised representation** learning methods: TS-TCC [Eldele et al., 2021], TST [Zerveas et al., 2021], TS2Vec [Yue et al., 2022], and T-Rep [Fraikin et al., 2024], (4) models **pre-trained** on multiple datasets: MOMENT [Goswami et al., 2024] and UniTS [Gao et al., 2024]. We compute the classification accuracy as metrics and compare the methods based on the statistics of accuracy and ranking. Details on benchmarking setups are included in Appendix A.

Table 1: Statistic and comparisons of the baselines and VQShape. The best case is marked with bold, the second best is marked with italic, and the third best is marked with underline. Some baselines fail are incompatible with some datasets which result in "N/A". For fair comparison, we report the statistics with and without "N/A". Complete results are presented in Table 5.

| | Classical | | Supervised | | | | | Unsupervised Representation | | | | Pre-trained | | |
| --- | --- | --- | --- | --- | --- | --- | --- | --- | --- | --- | --- | --- | --- | --- |
| | DTW | STRF | DLinear | Autoformer | FEDformer | PatchTST | TimesNet | TS-TCC | TST | T-Rep | TS2Vec | MOMENT | UniTS | VQShape |
| **Statistics with N/A** | | | | | | | | | | | | | | |
| Mean Accuracy | 0.648 | 0.660 | 0.635 | 0.570 | 0.612 | 0.669 | 0.710 | 0.682 | 0.630 | *0.719* | 0.712 | 0.686 | 0.629 | **0.723** |
| Median Accuracy | 0.711 | 0.679 | 0.673 | 0.553 | 0.586 | 0.756 | 0.797 | 0.753 | 0.620 | 0.804 | **0.812** | 0.759 | 0.684 | *0.810* |
| Mean Rank | 7.138 | 7.828 | 8.690 | 9.448 | 7.750 | 8.296 | *5.143* | 7.172 | 8.448 | 5.207 | **4.897** | 5.929 | 9.828 | 5.621 |
| Median Rank | 7.0 | 8.0 | 9.0 | 10.0 | 8.0 | 8.0 | 5.0 | 8.0 | 9.0 | *4.0* | **3.0** | 5.5 | 10.0 | 5.0 |
| Num. Top-1 | 3 | 2 | 1 | 0 | 1 | 0 | 1 | 3 | 1 | 4 | **6** | 5 | 0 | **6** |
| Num. Top-3 | 8 | 5 | 5 | 0 | 8 | 2 | 8 | 5 | 4 | *12* | **16** | 11 | 0 | 9 |
| Num. Win/Tie | 14 | 20 | 22 | 25 | 19 | 20 | 14 | 18 | 20 | 15 | 13 | 13 | 25 | - |
| Num. Lose | 15 | 9 | 7 | 4 | 9 | 7 | 14 | 11 | 9 | 14 | 16 | 15 | 4 | - |
| Wilcoxon p-value | 0.206 | 0.023 | 0.002 | 0.000 | 0.051 | 0.000 | 0.898 | 0.156 | 0.022 | 0.536 | 0.576 | 0.733 | 0.000 | - |
| **Statistics without N/A** | | | | | | | | | | | | | | |
| Mean Accuracy | 0.642 | 0.658 | 0.635 | 0.561 | 0.601 | 0.657 | 0.703 | 0.669 | 0.623 | *0.710* | 0.704 | 0.688 | 0.618 | **0.720** |
| Median Accuracy | 0.714 | 0.712 | 0.690 | 0.552 | 0.585 | 0.739 | 0.797 | 0.752 | 0.638 | *0.802* | 0.748 | 0.759 | 0.679 | **0.812** |
| Mean Rank | 7.231 | 7.692 | 8.923 | 9.462 | 7.731 | 8.346 | 5.308 | 7.538 | 8.462 | *5.192* | **5.038** | 5.885 | 10.115 | 5.538 |
| Median Rank | 6.5 | 7.5 | 9.5 | 9.5 | 8.0 | 8.5 | 5.000 | 8.0 | 9.5 | *4.5* | **3.0** | 5.5 | 10.5 | 4.5 |
| Num. Top-1 | 3 | 2 | 1 | 0 | 1 | 0 | 1 | 3 | 1 | 3 | **5** | 5 | 0 | **5** |
| Num. Top-3 | 7 | 5 | 5 | 0 | 8 | 2 | 7 | 4 | 4 | *10* | **14** | 10 | 0 | 8 |
| Num. Win/Tie | 13 | 18 | 20 | 22 | 17 | 20 | 13 | 17 | 19 | 14 | 12 | 12 | 23 | - |
| Num. Lose | 13 | 8 | 6 | 4 | 9 | 6 | 13 | 9 | 7 | 12 | 14 | 14 | 3 | - |
| Wilcoxon p-value | 0.187 | 0.036 | 0.001 | 0.001 | 0.111 | 0.001 | 0.803 | 0.094 | 0.036 | 0.653 | 0.696 | 1.000 | 0.000 | - |

**Results.** On multivariate TS classification tasks, for each TS sample $x_i \in \mathcal{X}$, we extract the token representations using VQShape and apply a learnable linear layer on this feature vector to predict the class label. Table 1 summarizes the performance of VQShape and baseline methods on the 29 UEA datasets. Complete results are presented in Table 5. From the results, we observe that the best-performing methods, including TimesNet, T-Rep, TS2Vec, MOMENT, and VQShape, have similar performance, and none of the methods has performance that is statistically significant compared to the others. Therefore, we conclude that VQShape achieves comparable performance to the state-of-the-art baselines while additionally providing interpretable pre-trained features.

**Frozen Pre-trained Representations.** Next, we compare the frozen pre-trained representations from three existing pre-trained models: MOMENT, UniTS, and VQShape. Since the pre-training dataset could have a dominant effect on the models, for fair comparisons, we reproduce MOMENT-small and UniTS by training them on the same datasets as VQShape. Table 2 summarizes the statistics of the pre-trained models on the 29 UEA datasets. Complete results are presented in Table 6. From the results, we observe that VQShape outperforms MOMENT and UniTS on most of the datasets and in overall statistics.

## 5.1 Generalizability

The experimental results presented above evaluate models pre-trained on the training splits of the UEA datasets and tested on the test splits of the same datasets, demonstrating the models' ability to generalize to unseen samples from the same domains. Besides in-domain generalizability, VQShape

Table 2: Comparison between three models pre-trained on all or a subset of the UEA datasets. The best cases are marked with bold. Complete results are presented in Table 6.

| Pre-trained on: | 29 datasets | | | 9 datasets | | |
|---|---|---|---|---|---|---|
| | MOMENT | UniTS | VQShape | MOMENT | UniTS | VQShape |
| Mean Accuracy | 0.697 | 0.581 | **0.723** | 0.697 | 0.559 | **0.723** |
| Median Accuracy | 0.736 | 0.649 | **0.810** | 0.733 | 0.649 | **0.792** |
| Mean Rank | 1.655 | 2.862 | **1.483** | 1.655 | 2.966 | **1.310** |
| Num. Top-1 | 13 | 0 | **16** | 11 | 0 | **20** |

and its codebook can also generalize to datasets and domains not observed during pre-training. To demonstrate cross-domain generalizability, we train another model using 9 datasets from the UEA archive that are commonly selected to train and evaluate deep learning models [Zerveas et al., 2021, Wu et al., 2023], and then evaluate it on all 29 datasets. The right half of Table 2 summarizes the performance of this model, compared with MOMENT and UniTS trained with the same setup. We observe that VQShape and MOMENT trained on fewer datasets result in similar but slightly worse performance, indicating that the representations learned by the models can generalize to unseen domains.

## 5.2 Interpretability

**Universal Codebook of Abstracted Shapes.** One of the most essential components of VQShape is the dataset-agnostic codebook that contains abstracted shapes. In Figure 6 of Appendix C.1, we decode all 512 codes in the codebook of VQShape to visualize their corresponding abstracted shapes. We observe that a large number of codes are decoded into similar shapes, which suggests that the codebook size can be further reduced. We then visualize the distribution of codes learned from pre-training (see Figure 7) which contains about 60 clusters. Inspired by this observation, we train a variant named VQShape-64 with codebook size $N^{\text{code}} = 64$. Figure 2 presents the decoded codebook of VQShape-64.

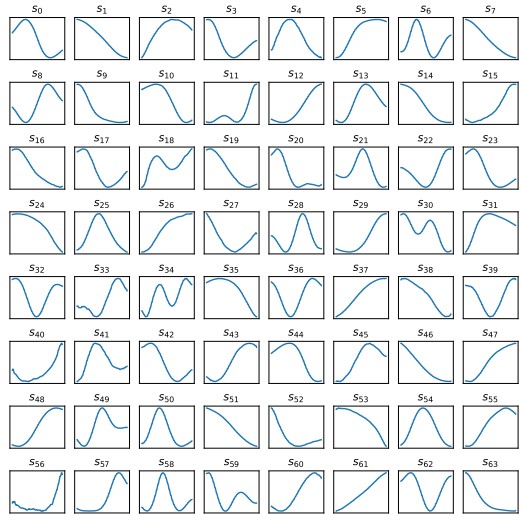

Figure 2: Visualization of the decoded codebook from VQShape-64.

**Interpretable Representations.** Overall, the encoding of VQShape can be interpreted as "TS $x$ is decomposed into (shape $z_1$ with offset $\mu_1$ and scale $\sigma_1$, at $t_1$ with length $l_1$), ...", and the decoding can be interpreted as "The composition of (shape $z_1$ with offset $\mu_1$ and scale $\sigma_1$, at $t_1$ with length $l_1$), ... becomes $\hat{x}$". Figure 3 includes an example of interpretable representations learned by VQShape. From visualizations, we can confirm that VQShape can learn abstracted shapes that capture shape-level information with various positions and scales.

**Discriminative Representations for Classification.** We further show that the interpretable representations produced by VQShape also capture discriminative patterns that distinguish different categories in classification tasks. Figure 4 visualizes the average code histogram for samples from two categories. From the feature maps, it is obvious that several codes have significant differences in frequency between the two categories; these serve as discriminative features in classification tasks. We decode and visualize their corresponding abstracted shapes. The intuition provided by the histogram features can be interpreted as: "Samples from the CW circle category usually contain shape $s_{61}$ in variate 1, and samples from the CCW circle category contain shape $s_{33}$ in variate 3, etc."

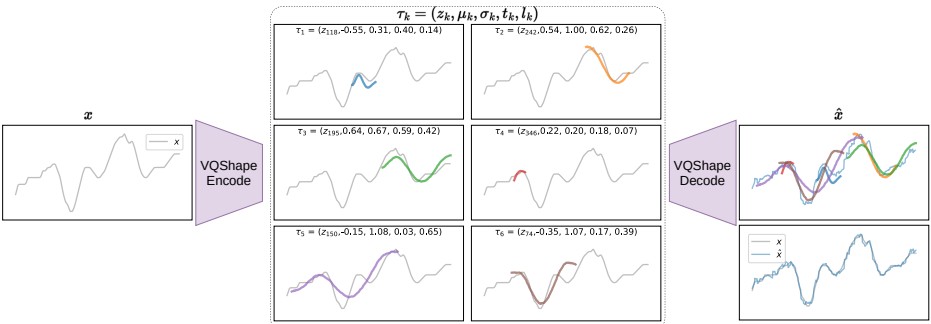

Figure 3: An example of abstracted shapes and their attributes (i.e., token representations) extracted by VQShape. For better presentation, we visualize 6 of the 64 shapes.

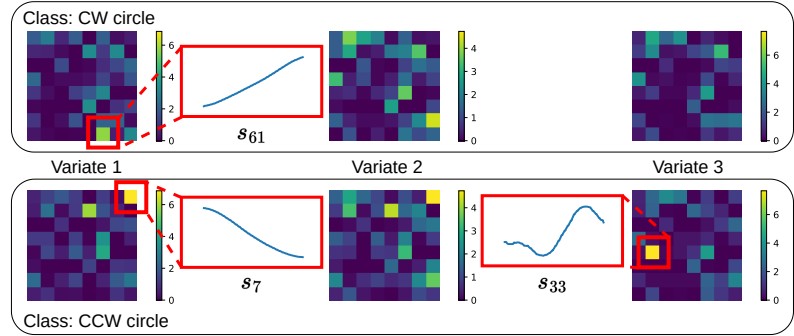

Figure 4: Example of how the code histogram representations provide discriminative features for classification. Histogram representations are obtained from VQShape-64. The histograms are averaged over samples of the two classes from the test split of the UWaveGestureLibrary dataset. The top and bottom rows represent samples labeled as "CW circle" and "CCW circle", respectively. Each column represent a variate (channel).

## 5.3 Ablation Study

**Representations and Codebook Scaling.** We compare the performance of classifiers trained with token or histogram representations from the same pre-trained VQShape models. We also study how the size of the codebook affects the quality of these representations. Based on the results presented in Figure 5, we conclude that token classifiers always outperform histogram classifiers since token representations are more expressive than histogram representations from the same pre-trained model.

Additionally, the performance of token classifiers increases steadily as codebook size increases, since tokens from larger codebooks could contain more details. However, for histogram representations, choosing the appropriate codebook size is essential; we observe that a codebook size of 64 results in the best performance. These results match our previous observation in Section 5.2 where the model with $N^{\text{code}} = 512$ results in only approximately 60 clusters of codes. A small codebook would learn codes that are too abstract to provide detailed representations, while a large codebook would learn similar codes that could lead to misleading features in the histograms. From

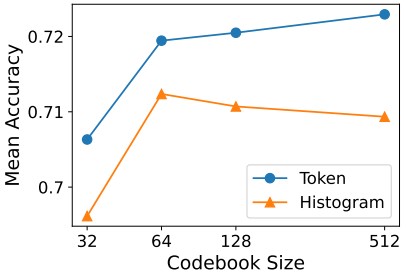

Figure 5: Mean accuracy of classifiers trained with token and histogram representations across different codebook sizes. Performance of the token classifiers improve with larger codebook, while performance of the histogram classifiers peak at codebook size 64 and decline with larger codebook.

the experimental results, we conclude that $N^{\text{code}} = 64$ strikes the best balance between abstraction and expressiveness for pre-training on the UEA datasets, producing the best histogram representations.

Note that within the scope of this paper, the number of code clusters remains a post-hoc discovery that can only be determined after pre-training. One could start with a large codebook size such as $N^{\text{code}} = 512$, visualize the distribution of codes, and re-train with the appropriate codebook size. Developing a mechanism to adjust the codebook size dynamically during training or inference would be a valuable direction for future work.

**Other Ablation Studies.** We additionally conduct two ablation studies: (1) setting $d^{\text{code}} = 32$ to compare high-dimensional and low-dimensional codes, and (2) setting $\lambda_s = 0$ to assess the value of introducing shape-level information to the VQ-VAE structure. Table 3 summarizes the statistics of these two cases and the default setting. The complete results are presented in Table 7 and Table 8. From the overall statistics, pre-training with $\lambda_s = 0$ leads to degraded performance, which indicates that learning shape-level abstractions through sub-

Table 3: Statistics of ablation cases on code dimension and shape reconstruction loss.

| Hyperparameter | | Value | | |
|---|---|---|---|---|
| Code dim | $d^{\text{code}}$ | 8 | 32 | 8 |
| Codebook size | $N^{\text{code}}$ | 512 | 512 | 512 |
| Shape loss | $\lambda_s$ | 1 | 1 | 0 |
| Mean Accuracy | Token | 0.723 | 0.721 | 0.708 |
| Median Accuracy | Token | 0.810 | 0.800 | 0.761 |
| Mean Accuracy | Histogram | 0.709 | 0.717 | 0.707 |
| Median Accuracy | Histogram | 0.762 | 0.810 | 0.747 |

sequence reconstruction introduces useful information for classification tasks. Using codes with $d^{\text{code}} = 32$ produces token classifiers with similar performance and better histogram classifiers. However, as stated in Section 3.2, we use low-dimensional code to create a bottleneck where the code should mainly contain the shape-level information. Using high-dimensional code may introduce additional information beyond the decoded shapes, which reduces interpretability. Therefore, the effect of using high-dimensional code would require extensive study.

# 6 Conclusion and Limitations

This paper introduces VQShape, a self-supervised pre-trained, interpretable, and generalizable model for TS analysis. Taking advantage of large pre-trained Transformers and ideas from shapelets, VQShape extracts a set of interpretable representations from TS data, composed of abstracted shape, offset, scale, position, and duration. Pre-trained and evaluated on multivariate classification datasets from the UEA archive, VQShape achieves comparable or even better performance than black-box models pre-trained on large datasets and with significantly more parameters, while providing additional interpretable representations. Furthermore, using VQShape, we present a codebook containing a set of abstracted shapes that generalize to various TS datasets, including unseen datasets.

In this paper, we do not term VQShape a foundation model for TS data, since the amount of pre-training data is still limited compared to other large pre-trained models such as MOMENT, and we are focusing only on classification tasks because the extracted shape tokens are mostly interpretable for classification. As presented in Table 6, based on the comparison between VQShape pre-trained on 29 and 9 datasets, simply adding more pre-training data may degrade performance on some datasets. This suggests that learning shape-level representations may not benefit from some types of data. For example, the BasicMotion datasets mostly contain signals with high-frequency sinusoidal components that do not contain meaningful shape-level information in short subsequences. Such datasets may "pollute" the pre-training of VQShape since the model will try to capture unnecessary high-frequency features. Therefore, if pre-trained on datasets at scale, additional pre-processing and input engineering should be included; however, we do not conduct explicit pre-processing in this paper. It would also be an important future step to develop interpretable frameworks for other TS analysis tasks, such as forecasting, imputation, and anomaly detection, using the interpretable tokens extracted by VQShape.

# Acknowledgments

This work was supported in part by IBM through the IBM-Rensselaer Future of Computing Research Collaboration. We thank Dr. Eamonn Keogh for his valuable comments and suggestions for revision. We also appreciate the UEA team's effort in creating and publicly sharing the benchmark datasets.

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

# A Experiment Setup

**Environment.** VQShape is implemented using Python 3.11.8 and PyTorch 2.2.2. The pre-training and evaluations are conducted on a machine with Intel Core i7-11700k CPU, 32GB of RAM, and a Nvidia RTX 4090 24GB GPU.

Table 4: Information of the 30 benchmark datasets from the UEA archive.

| Dataset | Train Size $N_{\text{train}}$ | Test Size $N_{\text{test}}$ | Variables $M$ | Length $T$ | Categories $C$ | Type |
|---|---|---|---|---|---|---|
| ArticularyWordRecognition | 275 | 300 | 9 | 144 | 25 | MOTION |
| AtrialFibrillation | 15 | 15 | 2 | 640 | 3 | ECG |
| BasicMotions | 40 | 40 | 6 | 100 | 4 | HAR |
| CharacterTrajectories | 1422 | 1436 | 3 | 119 | 20 | MOTION |
| Cricket | 108 | 72 | 6 | 1197 | 12 | HAR |
| DuckDuckGeese | 50 | 50 | 1345 | 270 | 5 | AUDIO |
| ERing | 30 | 270 | 4 | 65 | 6 | HAR |
| EigenWorms | 128 | 131 | 6 | 17948 | 5 | MOTION |
| Epilepsy | 137 | 138 | 3 | 206 | 4 | HAR |
| EthanolConcentration | 261 | 263 | 3 | 1751 | 4 | OTHER |
| FaceDetection | 5890 | 3524 | 144 | 62 | 2 | EEG |
| FingerMovements | 316 | 100 | 28 | 50 | 2 | EEG |
| HandMovementDirection | 160 | 74 | 10 | 400 | 4 | EEG |
| Handwriting | 150 | 580 | 3 | 52 | 26 | HAR |
| Heartbeat | 204 | 205 | 61 | 405 | 2 | AUDIO |
| InsectWingbeat | 25000 | 25000 | 200 | 22 | 10 | AUDIO |
| JapaneseVowels | 270 | 370 | 12 | 29 | 9 | AUDIO |
| LSST | 2459 | 2466 | 6 | 36 | 14 | OTHER |
| Libras | 180 | 180 | 2 | 45 | 15 | HAR |
| MotorImagery | 278 | 100 | 64 | 3000 | 2 | EEG |
| NATOPS | 180 | 180 | 24 | 51 | 6 | HAR |
| PEMS-SF | 267 | 173 | 263 | 144 | 7 | MISC |
| PenDigits | 7494 | 3498 | 2 | 8 | 10 | MOTION |
| PhonemeSpectra | 3315 | 3353 | 11 | 217 | 39 | SOUND |
| RacketSports | 151 | 152 | 6 | 30 | 4 | HAR |
| SelfRegulationSCP1 | 268 | 293 | 6 | 896 | 2 | EEG |
| SelfRegulationSCP2 | 200 | 180 | 7 | 1152 | 2 | EEG |
| SpokenArabicDigits | 6599 | 2199 | 13 | 93 | 10 | SPEECH |
| StandWalkJump | 12 | 15 | 4 | 2500 | 3 | ECG |
| UWaveGestureLibrary | 120 | 320 | 3 | 315 | 8 | HAR |

**Classification tasks.** Taking frozen pre-trained representations from VQShape, we learn a linear classifier to make predictions. When training the linear classifier, we found that token representations work better with regularization of L2 on classifier weights, while histogram representations are more compatible with dropout on features. Therefore, to avoid overfitting and obtain the optimal performance, we tune both the L2 regularization weight (or weight decay) and the dropout rate.

When learning the linear classifier, we repeat the experiment with five random seeds and report the average accuracy in Table 5. The standard deviations of the five runs are included in Table 7. Note that we exclude the `InsectWingbeat` dataset since the dataset contains inconsistent and very short TS samples such as $T = 1$. Considering that the dataset has significantly more samples and channels than other datasets, the high volume of such short samples may have a negative effect on our method since the short TS do not contain any meaningful shape-level features.

**Baseline Results.** For baseline results presented in Table 1 and Table 5, we reproduce STRF with the Aeon-Tookit[2] and Scikit-learn packages. We reproduce DLinear, Autoformer, FEDformer, PatchTST, and TimesNet using implementation from Wu et al. [2023] [3]. Results of DTW, TS-TCC, TST, and TS2Vec are obtained from the TS2Vec paper [Yue et al., 2022]. Results of MOMENT

---

[2]The Aeon-Tookit is available at `www.aeon-toolkit.org`.

[3]The implementations are available at `https://github.com/thuml/Time-Series-Library`.

[Goswami et al., 2024] and T-Rep Fraikin et al. [2024] are obtained from the original papers. For results of UniTS [Gao et al., 2024], since the full results on the UEA datasets are not reported in the original paper, we follow the official implementation to reproduce the results using a pre-trained checkpoint [4] and prompt learning.

**Benchmarking Frozen Pre-trained Models.** To obtain the results in Table 2 and Table 6, we train MOMENT and UniTS from scratch using the official implementations [5]. Note that the way MOMENT, UniTS, and VQShape produce predictions can be different. MOMENT and VQShape provide frozen representations and learn a separate classifier, whereas UniTS learns additional prompt tokens and classification heads. Therefore, we follow the default procedure and the official implementation to evaluate UniTS. We evaluate MOMENT and VQShape under the exact same procedure of learning linear classifiers with regularization tuning.

Note that the key difference between results in Table 5 and Table 6 is that MOMENT and UniTS in Table 5 are pre-trained on a significantly larger volume of data (which includes the forecasting datasets), while the three models are only pre-trained on the UEA datasets in Table 6.

# B  Full Experiment Results

Table 5: Full benchmark results on the 29 UEA datasets. For each dataset, the best case is marked with bold, the second best is marked with italic, and the third best is marked with underline.

| | Classical | | Supervised | | | | | Unsupervised Representation | | | | Pre-trained | | |
|---|---|---|---|---|---|---|---|---|---|---|---|---|---|---|
| Dataset | DTW | STRF | DLinear | Autoformer | FEDformer | PatchTST | TimesNet | TS-TCC | TST | T-Rep | TS2Vec | MOMENT | UniTS | VQShape |
| ArticularyWordRecognition | 0.987 | 0.917 | 0.963 | 0.567 | 0.593 | 0.927 | 0.973 | 0.953 | 0.977 | 0.968 | 0.987 | 0.990 | 0.927 | 0.987 |
| AtrialFibrillation | 0.200 | 0.267 | 0.200 | 0.333 | 0.467 | 0.333 | 0.333 | 0.267 | 0.067 | 0.354 | 0.200 | 0.200 | 0.133 | 0.520 |
| BasicMotions | 0.975 | 0.925 | 0.850 | 0.550 | 0.800 | 0.700 | 0.975 | 1.000 | 0.975 | 1.000 | 0.975 | 1.000 | 0.600 | 0.910 |
| CharacterTrajectories | 0.989 | 0.849 | 0.986 | 0.894 | 0.914 | 0.976 | 0.987 | 0.985 | 0.975 | 0.989 | 0.995 | N/A | 0.935 | 0.969 |
| Cricket | 1.000 | 0.944 | 0.861 | 0.194 | 0.389 | 0.889 | 0.903 | 0.917 | 1.000 | 0.958 | 0.972 | 1.000 | 0.958 | 0.978 |
| DuckDuckGeese | 0.600 | 0.380 | 0.500 | 0.340 | 0.260 | 0.220 | 0.580 | 0.380 | 0.620 | 0.457 | 0.680 | 0.600 | 0.320 | 0.360 |
| ERing | 0.133 | 0.889 | 0.844 | 0.715 | 0.733 | 0.937 | 0.927 | 0.904 | 0.874 | 0.943 | 0.874 | 0.959 | 0.830 | 0.960 |
| EigenWorms | 0.618 | 0.672 | 0.321 | 0.504 | N/A | N/A | N/A | 0.779 | 0.618 | 0.884 | 0.847 | 0.809 | 0.710 | 0.603 |
| Epilepsy | 0.964 | 0.978 | 0.565 | 0.746 | 0.804 | 0.913 | 0.877 | 0.957 | 0.949 | 0.970 | 0.964 | 0.993 | 0.942 | 0.893 |
| EthanolConcentration | 0.262 | 0.677 | 0.270 | 0.255 | 0.281 | 0.259 | 0.285 | 0.285 | 0.262 | 0.333 | 0.308 | 0.357 | 0.259 | 0.325 |
| FaceDetection | 0.529 | 0.567 | 0.673 | 0.585 | 0.688 | 0.668 | 0.677 | 0.544 | 0.534 | 0.581 | 0.501 | 0.633 | 0.549 | 0.653 |
| FingerMovements | 0.530 | 0.500 | 0.570 | 0.550 | 0.540 | 0.580 | 0.530 | 0.460 | 0.560 | 0.495 | 0.480 | 0.490 | 0.520 | 0.642 |
| HandMovementDirection | 0.231 | 0.419 | 0.662 | 0.378 | 0.365 | 0.514 | 0.595 | 0.243 | 0.243 | 0.536 | 0.338 | 0.324 | 0.365 | 0.546 |
| Handwriting | 0.286 | 0.104 | 0.189 | 0.147 | 0.185 | 0.251 | 0.311 | 0.498 | 0.225 | 0.414 | 0.515 | 0.308 | 0.137 | 0.270 |
| Heartbeat | 0.717 | 0.746 | 0.707 | 0.737 | 0.741 | 0.722 | 0.732 | 0.751 | 0.746 | 0.725 | 0.683 | 0.722 | 0.673 | 0.663 |
| JapaneseVowels | 0.949 | 0.676 | 0.949 | 0.951 | 0.968 | 0.935 | 0.954 | 0.930 | 0.978 | 0.962 | 0.984 | 0.716 | 0.822 | 0.945 |
| LSST | 0.551 | 0.491 | 0.317 | 0.418 | 0.363 | 0.519 | 0.382 | 0.474 | 0.408 | 0.526 | 0.537 | 0.411 | 0.492 | 0.511 |
| Libras | 0.870 | 0.817 | 0.506 | 0.767 | 0.867 | 0.761 | 0.761 | 0.822 | 0.656 | 0.829 | 0.867 | 0.850 | 0.750 | 0.814 |
| MotorImagery | 0.500 | 0.510 | 0.600 | 0.560 | 0.580 | N/A | 0.610 | 0.610 | 0.500 | 0.495 | 0.510 | 0.500 | 0.540 | 0.680 |
| NATOPS | 0.883 | 0.794 | 0.917 | 0.744 | 0.828 | 0.756 | 0.833 | 0.822 | 0.850 | 0.804 | 0.928 | 0.828 | 0.756 | 0.810 |
| PEMS-SF | 0.711 | 0.925 | 0.809 | 0.838 | 0.867 | 0.809 | 0.844 | 0.734 | 0.740 | 0.800 | 0.682 | 0.896 | 0.844 | 0.865 |
| PenDigits | 0.977 | 0.855 | 0.851 | 0.982 | 0.985 | 0.974 | 0.984 | 0.974 | 0.560 | 0.971 | 0.989 | 0.972 | 0.894 | 0.973 |
| PhonemeSpectra | 0.151 | 0.155 | 0.069 | 0.086 | 0.099 | 0.081 | 0.146 | 0.252 | 0.085 | 0.232 | 0.233 | 0.233 | 0.119 | 0.087 |
| RacketSports | 0.803 | 0.842 | 0.730 | 0.822 | 0.822 | 0.757 | 0.855 | 0.816 | 0.809 | 0.883 | 0.855 | 0.796 | 0.684 | 0.851 |
| SelfRegulationSCP1 | 0.775 | 0.846 | 0.881 | 0.553 | 0.577 | 0.795 | 0.908 | 0.823 | 0.754 | 0.819 | 0.812 | 0.840 | 0.795 | 0.904 |
| SelfRegulationSCP2 | 0.539 | 0.489 | 0.528 | 0.544 | 0.522 | 0.506 | 0.539 | 0.533 | 0.550 | 0.591 | 0.578 | 0.478 | 0.528 | 0.596 |
| SpokenArabicDigits | 0.963 | 0.679 | 0.965 | 0.985 | 0.988 | 0.977 | 0.988 | 0.970 | 0.923 | 0.994 | 0.988 | 0.981 | 0.924 | 0.976 |
| StandWalkJump | 0.200 | 0.467 | 0.333 | 0.333 | 0.467 | 0.467 | 0.533 | 0.333 | 0.267 | 0.441 | 0.467 | 0.400 | 0.400 | 0.787 |
| UWaveGestureLibrary | 0.903 | 0.762 | 0.812 | 0.466 | 0.434 | 0.828 | 0.863 | 0.753 | 0.575 | 0.885 | 0.906 | 0.909 | 0.838 | 0.888 |

---

[4] Implementation and checkpoints of UniTS are available at `https://github.com/mims-harvard/UniTS`.

[5] Implementation and pre-training for MOMENT are available at `https://github.com/moment-timeseries-foundation-model/moment-research`.

Table 6: Full results of three models pre-trained on the UEA datasets. The best case for models pre-trained on 29 datasets is marked with bold, and the best case for models pre-trained on 9 datasets is marked with underline. The 9 datasets are marked with †.

| Pre-trained on: | 29 datasets | | | 9 datasets | | |
| --- | --- | --- | --- | --- | --- | --- |
| Dataset | MOMENT | UniTS | VQShape | MOMENT | UniTS | VQShape |
| ArticularyWordRecognition | **0.990** | 0.860 | 0.987 | 0.987 | 0.880 | 0.990 |
| AtrialFibrillation | **0.533** | 0.400 | 0.520 | 0.480 | 0.400 | 0.480 |
| BasicMotions | 0.760 | 0.775 | **0.910** | 0.790 | 0.775 | 0.950 |
| CharacterTrajectories | **0.982** | 0.723 | 0.969 | 0.982 | 0.735 | 0.966 |
| Cricket | **0.986** | 0.944 | 0.978 | 0.989 | 0.819 | 0.989 |
| DuckDuckGeese | **0.464** | 0.400 | 0.360 | 0.384 | 0.240 | 0.396 |
| ERing | 0.907 | 0.700 | **0.960** | 0.937 | 0.756 | 0.976 |
| EigenWorms | **0.663** | 0.466 | 0.603 | 0.667 | 0.389 | 0.583 |
| Epilepsy | **0.987** | 0.826 | 0.893 | 0.983 | 0.819 | 0.877 |
| EthanolConcentration † | **0.414** | 0.213 | 0.325 | 0.424 | 0.247 | 0.303 |
| FaceDetection † | 0.597 | 0.500 | **0.653** | 0.601 | 0.529 | 0.649 |
| FingerMovements | 0.630 | 0.510 | **0.642** | 0.630 | 0.490 | 0.638 |
| HandMovementDirection | 0.381 | 0.257 | **0.546** | 0.408 | 0.216 | 0.508 |
| Handwriting † | 0.225 | 0.087 | **0.270** | 0.245 | 0.153 | 0.280 |
| Heartbeat † | **0.744** | 0.649 | 0.663 | 0.733 | 0.649 | 0.661 |
| JapaneseVowels † | 0.706 | 0.843 | **0.945** | 0.709 | 0.824 | 0.952 |
| LSST | 0.429 | 0.361 | **0.511** | 0.423 | 0.415 | 0.510 |
| Libras | **0.908** | 0.633 | 0.814 | 0.879 | 0.556 | 0.813 |
| MotorImagery | 0.642 | 0.510 | **0.680** | 0.682 | 0.460 | 0.672 |
| NATOPS | **0.860** | 0.711 | 0.810 | 0.862 | 0.661 | 0.792 |
| PEMS-SF † | **0.875** | 0.821 | 0.865 | 0.891 | 0.861 | 0.889 |
| PenDigits | 0.965 | 0.819 | **0.973** | 0.965 | 0.669 | 0.970 |
| PhonemeSpectra | **0.090** | 0.070 | 0.087 | 0.071 | 0.069 | 0.085 |
| RacketSports | 0.736 | 0.671 | **0.851** | 0.763 | 0.678 | 0.882 |
| SelfRegulationSCP1 † | 0.829 | 0.652 | **0.904** | 0.821 | 0.717 | 0.898 |
| SelfRegulationSCP2 † | 0.576 | 0.506 | **0.596** | 0.588 | 0.417 | 0.624 |
| SpokenArabicDigits | 0.971 | 0.751 | **0.976** | 0.963 | 0.769 | 0.975 |
| StandWalkJump | 0.493 | 0.533 | **0.787** | 0.507 | 0.333 | 0.747 |
| UWaveGestureLibrary † | 0.871 | 0.656 | **0.888** | 0.846 | 0.688 | 0.902 |

Table 7: Full results of VQShape variants with token classifiers. For a frozen pre-trained model, the mean and standard deviation of accuracies over five randomly initialized linear classifiers are reported.

| Hyperparameter | | Variants of VQShape | | | | | |
|---|---|---|---|---|---|---|---|
| Codebook Size | $N^{code}$: | 32 | 64 | 128 | 512 | 512 | 512 |
| Code Dim. | $d^{code}$: | 8 | 8 | 8 | 8 | 8 | 32 |
| Shape Loss Weight | $\lambda_s$: | 1 | 1 | 1 | 1 | 0 | 1 |
| Representation: | | Token | Token | Token | Token | Token | Token |
| ArticularyWordRecognition | | $0.979 \pm 0.032$ | $0.988 \pm 0.021$ | $0.990 \pm 0.024$ | $0.987 \pm 0.018$ | $0.981 \pm 0.030$ | $0.991 \pm 0.025$ |
| AtrialFibrillation | | $0.547 \pm 0.073$ | $0.520 \pm 0.191$ | $0.573 \pm 0.068$ | $0.520 \pm 0.068$ | $0.547 \pm 0.090$ | $0.653 \pm 0.107$ |
| BasicMotions | | $0.900 \pm 0.048$ | $0.925 \pm 0.037$ | $0.940 \pm 0.029$ | $0.910 \pm 0.044$ | $0.950 \pm 0.053$ | $0.960 \pm 0.046$ |
| CharacterTrajectories | | $0.960 \pm 0.019$ | $0.965 \pm 0.029$ | $0.963 \pm 0.027$ | $0.969 \pm 0.030$ | $0.958 \pm 0.035$ | $0.958 \pm 0.037$ |
| Cricket | | $0.978 \pm 0.022$ | $0.981 \pm 0.019$ | $0.986 \pm 0.010$ | $0.978 \pm 0.016$ | $0.986 \pm 0.023$ | $0.986 \pm 0.032$ |
| DuckDuckGeese | | $0.384 \pm 0.047$ | $0.400 \pm 0.039$ | $0.460 \pm 0.060$ | $0.360 \pm 0.045$ | $0.372 \pm 0.043$ | $0.408 \pm 0.046$ |
| ERing | | $0.966 \pm 0.016$ | $0.979 \pm 0.006$ | $0.984 \pm 0.010$ | $0.960 \pm 0.011$ | $0.761 \pm 0.022$ | $0.973 \pm 0.016$ |
| EigenWorms | | $0.589 \pm 0.034$ | $0.597 \pm 0.036$ | $0.569 \pm 0.031$ | $0.603 \pm 0.035$ | $0.608 \pm 0.035$ | $0.608 \pm 0.034$ |
| Epilepsy | | $0.730 \pm 0.023$ | $0.800 \pm 0.031$ | $0.841 \pm 0.020$ | $0.893 \pm 0.031$ | $0.803 \pm 0.029$ | $0.817 \pm 0.030$ |
| EthanolConcentration | | $0.313 \pm 0.020$ | $0.329 \pm 0.018$ | $0.317 \pm 0.019$ | $0.325 \pm 0.020$ | $0.335 \pm 0.018$ | $0.313 \pm 0.018$ |
| FaceDetection | | $0.640 \pm 0.006$ | $0.657 \pm 0.007$ | $0.639 \pm 0.013$ | $0.653 \pm 0.011$ | $0.672 \pm 0.007$ | $0.651 \pm 0.008$ |
| FingerMovements | | $0.640 \pm 0.029$ | $0.646 \pm 0.039$ | $0.630 \pm 0.029$ | $0.642 \pm 0.025$ | $0.644 \pm 0.032$ | $0.630 \pm 0.025$ |
| HandMovementDirection | | $0.438 \pm 0.046$ | $0.478 \pm 0.031$ | $0.459 \pm 0.048$ | $0.546 \pm 0.049$ | $0.565 \pm 0.036$ | $0.457 \pm 0.041$ |
| Handwriting | | $0.234 \pm 0.009$ | $0.278 \pm 0.016$ | $0.284 \pm 0.014$ | $0.270 \pm 0.012$ | $0.267 \pm 0.010$ | $0.255 \pm 0.013$ |
| Heartbeat | | $0.635 \pm 0.032$ | $0.633 \pm 0.021$ | $0.609 \pm 0.030$ | $0.663 \pm 0.029$ | $0.602 \pm 0.028$ | $0.621 \pm 0.024$ |
| JapaneseVowels | | $0.950 \pm 0.022$ | $0.950 \pm 0.019$ | $0.955 \pm 0.014$ | $0.945 \pm 0.014$ | $0.937 \pm 0.017$ | $0.941 \pm 0.029$ |
| LSST | | $0.476 \pm 0.019$ | $0.483 \pm 0.013$ | $0.509 \pm 0.012$ | $0.511 \pm 0.011$ | $0.502 \pm 0.015$ | $0.495 \pm 0.015$ |
| Libras | | $0.809 \pm 0.016$ | $0.804 \pm 0.020$ | $0.801 \pm 0.013$ | $0.814 \pm 0.029$ | $0.769 \pm 0.013$ | $0.804 \pm 0.019$ |
| MotorImagery | | $0.678 \pm 0.046$ | $0.670 \pm 0.037$ | $0.656 \pm 0.029$ | $0.680 \pm 0.039$ | $0.668 \pm 0.024$ | $0.690 \pm 0.046$ |
| NATOPS | | $0.781 \pm 0.020$ | $0.820 \pm 0.041$ | $0.800 \pm 0.023$ | $0.810 \pm 0.021$ | $0.790 \pm 0.025$ | $0.800 \pm 0.022$ |
| PEMS-SF | | $0.842 \pm 0.042$ | $0.876 \pm 0.038$ | $0.872 \pm 0.059$ | $0.865 \pm 0.044$ | $0.840 \pm 0.032$ | $0.837 \pm 0.041$ |
| PenDigits | | $0.967 \pm 0.025$ | $0.969 \pm 0.032$ | $0.970 \pm 0.029$ | $0.973 \pm 0.038$ | $0.949 \pm 0.033$ | $0.968 \pm 0.044$ |
| PhonemeSpectra | | $0.083 \pm 0.003$ | $0.077 \pm 0.003$ | $0.085 \pm 0.003$ | $0.087 \pm 0.003$ | $0.107 \pm 0.003$ | $0.092 \pm 0.003$ |
| RacketSports | | $0.864 \pm 0.026$ | $0.855 \pm 0.028$ | $0.888 \pm 0.015$ | $0.851 \pm 0.019$ | $0.841 \pm 0.013$ | $0.872 \pm 0.029$ |
| SelfRegulationSCP1 | | $0.881 \pm 0.023$ | $0.904 \pm 0.026$ | $0.887 \pm 0.025$ | $0.904 \pm 0.039$ | $0.904 \pm 0.014$ | $0.880 \pm 0.020$ |
| SelfRegulationSCP2 | | $0.597 \pm 0.024$ | $0.593 \pm 0.027$ | $0.606 \pm 0.034$ | $0.596 \pm 0.018$ | $0.587 \pm 0.025$ | $0.600 \pm 0.029$ |
| SpokenArabicDigits | | $0.971 \pm 0.020$ | $0.974 \pm 0.020$ | $0.977 \pm 0.019$ | $0.976 \pm 0.021$ | $0.977 \pm 0.028$ | $0.976 \pm 0.023$ |
| StandWalkJump | | $0.760 \pm 0.108$ | $0.813 \pm 0.103$ | $0.733 \pm 0.100$ | $0.787 \pm 0.131$ | $0.760 \pm 0.112$ | $0.787 \pm 0.126$ |
| UWaveGestureLibrary | | $0.893 \pm 0.017$ | $0.898 \pm 0.009$ | $0.912 \pm 0.019$ | $0.887 \pm 0.012$ | $0.846 \pm 0.010$ | $0.891 \pm 0.013$ |

Table 8: Full results of VQShape variants with histogram classifiers. For a frozen pre-trained model, the mean and standard deviation of accuracies over five randomly initialized linear classifiers are reported.

| Hyperparameter | | Variants of VQShape | | | | | |
|---|---|---|---|---|---|---|---|
| Codebook Size | $N^{\text{code}}$: | 32 | 64 | 128 | 512 | 512 | 512 |
| Code Dim. | $d^{\text{code}}$: | 8 | 8 | 8 | 8 | 8 | 32 |
| Shape Loss Weight | $\lambda_s$: | 1 | 1 | 1 | 1 | 0 | 1 |
| Representation: | | Histogram | Histogram | Histogram | Histogram | Histogram | Histogram |
| ArticularyWordRecognition | | $0.983 \pm 0.056$ | $0.996 \pm 0.050$ | $0.993 \pm 0.038$ | $0.991 \pm 0.034$ | $0.986 \pm 0.043$ | $0.988 \pm 0.030$ |
| AtrialFibrillation | | $0.560 \pm 0.084$ | $0.600 \pm 0.090$ | $0.533 \pm 0.144$ | $0.573 \pm 0.100$ | $0.520 \pm 0.103$ | $0.507 \pm 0.112$ |
| BasicMotions | | $1.000 \pm 0.041$ | $1.000 \pm 0.019$ | $0.985 \pm 0.012$ | $0.955 \pm 0.029$ | $0.930 \pm 0.025$ | $1.000 \pm 0.016$ |
| CharacterTrajectories | | $0.888 \pm 0.033$ | $0.908 \pm 0.030$ | $0.931 \pm 0.034$ | $0.942 \pm 0.040$ | $0.928 \pm 0.037$ | $0.938 \pm 0.036$ |
| Cricket | | $0.967 \pm 0.026$ | $0.986 \pm 0.033$ | $1.000 \pm 0.022$ | $0.986 \pm 0.024$ | $0.975 \pm 0.015$ | $0.978 \pm 0.034$ |
| DuckDuckGeese | | $0.428 \pm 0.045$ | $0.468 \pm 0.054$ | $0.440 \pm 0.039$ | $0.396 \pm 0.032$ | $0.360 \pm 0.042$ | $0.396 \pm 0.048$ |
| ERing | | $0.929 \pm 0.064$ | $0.956 \pm 0.019$ | $0.856 \pm 0.018$ | $0.927 \pm 0.013$ | $0.935 \pm 0.017$ | $0.928 \pm 0.016$ |
| EigenWorms | | $0.612 \pm 0.027$ | $0.571 \pm 0.023$ | $0.553 \pm 0.017$ | $0.669 \pm 0.018$ | $0.747 \pm 0.016$ | $0.602 \pm 0.018$ |
| Epilepsy | | $0.929 \pm 0.040$ | $0.959 \pm 0.027$ | $0.964 \pm 0.033$ | $0.970 \pm 0.026$ | $0.858 \pm 0.043$ | $1.000 \pm 0.039$ |
| EthanolConcentration | | $0.301 \pm 0.012$ | $0.313 \pm 0.018$ | $0.308 \pm 0.014$ | $0.319 \pm 0.017$ | $0.319 \pm 0.020$ | $0.310 \pm 0.015$ |
| FaceDetection | | $0.567 \pm 0.009$ | $0.567 \pm 0.010$ | $0.578 \pm 0.008$ | $0.584 \pm 0.006$ | $0.655 \pm 0.013$ | $0.601 \pm 0.009$ |
| FingerMovements | | $0.598 \pm 0.028$ | $0.598 \pm 0.033$ | $0.600 \pm 0.036$ | $0.600 \pm 0.024$ | $0.600 \pm 0.026$ | $0.600 \pm 0.028$ |
| HandMovementDirection | | $0.441 \pm 0.037$ | $0.449 \pm 0.028$ | $0.438 \pm 0.030$ | $0.432 \pm 0.045$ | $0.481 \pm 0.031$ | $0.427 \pm 0.046$ |
| Handwriting | | $0.156 \pm 0.009$ | $0.212 \pm 0.010$ | $0.196 \pm 0.007$ | $0.148 \pm 0.008$ | $0.235 \pm 0.008$ | $0.206 \pm 0.010$ |
| Heartbeat | | $0.736 \pm 0.007$ | $0.743 \pm 0.006$ | $0.741 \pm 0.011$ | $0.739 \pm 0.011$ | $0.730 \pm 0.008$ | $0.743 \pm 0.008$ |
| JapaneseVowels | | $0.939 \pm 0.048$ | $0.948 \pm 0.043$ | $0.945 \pm 0.050$ | $0.926 \pm 0.048$ | $0.936 \pm 0.044$ | $0.942 \pm 0.039$ |
| LSST | | $0.515 \pm 0.034$ | $0.521 \pm 0.035$ | $0.542 \pm 0.029$ | $0.515 \pm 0.025$ | $0.516 \pm 0.025$ | $0.556 \pm 0.018$ |
| Libras | | $0.768 \pm 0.020$ | $0.804 \pm 0.029$ | $0.811 \pm 0.023$ | $0.818 \pm 0.020$ | $0.730 \pm 0.021$ | $0.850 \pm 0.024$ |
| MotorImagery | | $0.628 \pm 0.024$ | $0.672 \pm 0.022$ | $0.650 \pm 0.028$ | $0.662 \pm 0.030$ | $0.626 \pm 0.042$ | $0.678 \pm 0.029$ |
| NATOPS | | $0.847 \pm 0.020$ | $0.844 \pm 0.022$ | $0.886 \pm 0.020$ | $0.808 \pm 0.016$ | $0.790 \pm 0.022$ | $0.864 \pm 0.020$ |
| PEMS-SF | | $0.839 \pm 0.050$ | $0.864 \pm 0.036$ | $0.853 \pm 0.036$ | $0.846 \pm 0.066$ | $0.810 \pm 0.034$ | $0.824 \pm 0.062$ |
| PenDigits | | $0.898 \pm 0.063$ | $0.942 \pm 0.075$ | $0.949 \pm 0.067$ | $0.969 \pm 0.038$ | $0.915 \pm 0.048$ | $0.958 \pm 0.030$ |
| PhonemeSpectra | | $0.116 \pm 0.003$ | $0.134 \pm 0.006$ | $0.132 \pm 0.005$ | $0.143 \pm 0.004$ | $0.104 \pm 0.005$ | $0.150 \pm 0.004$ |
| RacketSports | | $0.774 \pm 0.046$ | $0.879 \pm 0.055$ | $0.883 \pm 0.038$ | $0.838 \pm 0.027$ | $0.861 \pm 0.060$ | $0.832 \pm 0.027$ |
| SelfRegulationSCP1 | | $0.731 \pm 0.027$ | $0.754 \pm 0.027$ | $0.788 \pm 0.022$ | $0.762 \pm 0.030$ | $0.881 \pm 0.015$ | $0.810 \pm 0.028$ |
| SelfRegulationSCP2 | | $0.576 \pm 0.027$ | $0.588 \pm 0.026$ | $0.577 \pm 0.023$ | $0.572 \pm 0.022$ | $0.617 \pm 0.025$ | $0.583 \pm 0.028$ |
| SpokenArabicDigits | | $0.942 \pm 0.032$ | $0.959 \pm 0.044$ | $0.961 \pm 0.048$ | $0.969 \pm 0.061$ | $0.975 \pm 0.045$ | $0.969 \pm 0.051$ |
| StandWalkJump | | $0.680 \pm 0.116$ | $0.560 \pm 0.060$ | $0.613 \pm 0.094$ | $0.613 \pm 0.068$ | $0.653 \pm 0.073$ | $0.640 \pm 0.073$ |
| UWaveGestureLibrary | | $0.841 \pm 0.033$ | $0.864 \pm 0.036$ | $0.904 \pm 0.016$ | $0.899 \pm 0.030$ | $0.821 \pm 0.014$ | $0.923 \pm 0.026$ |

# C Additional Visualizations

## C.1 Visualization of abstracted shapes in the codebook

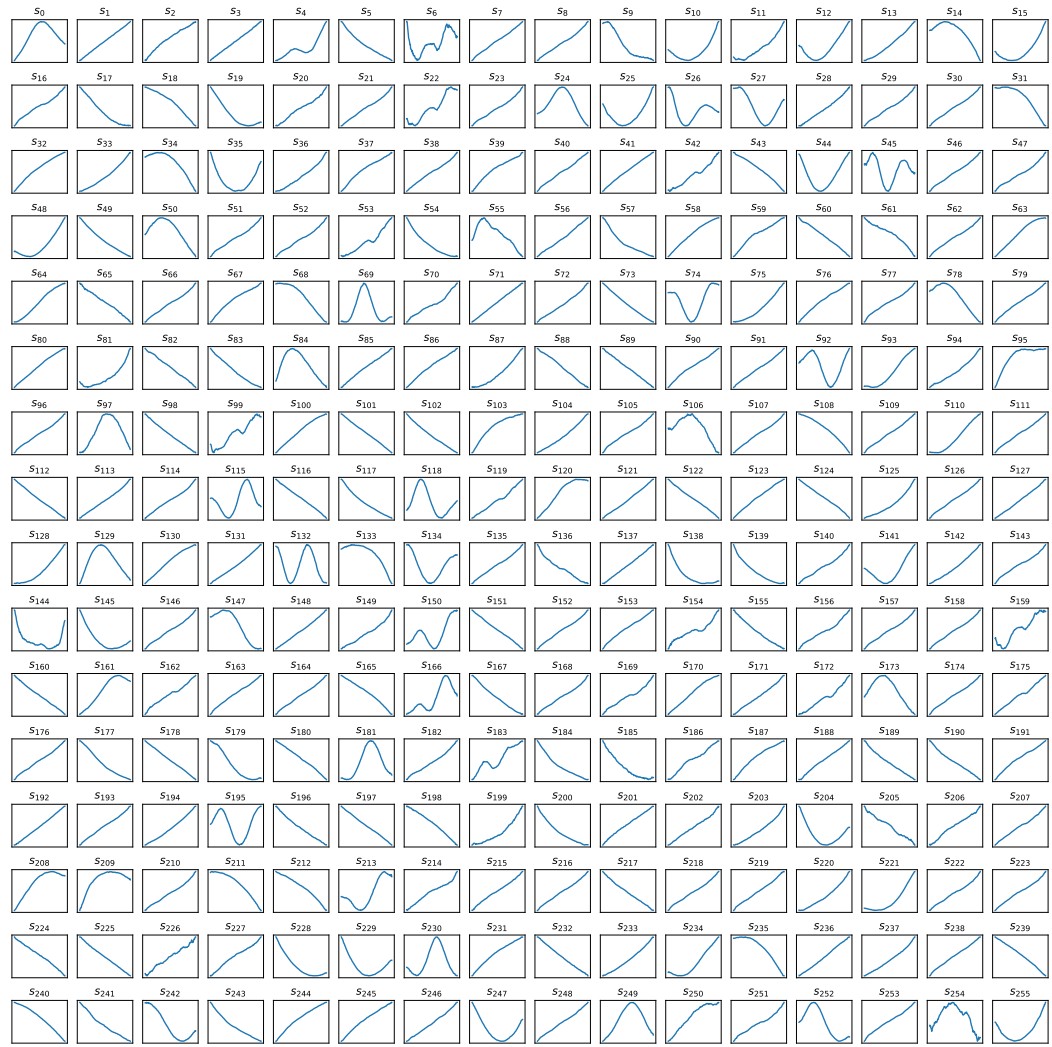

Figure 6: Visualizations of the abstracted shapes decoded from the codebook of VQShape

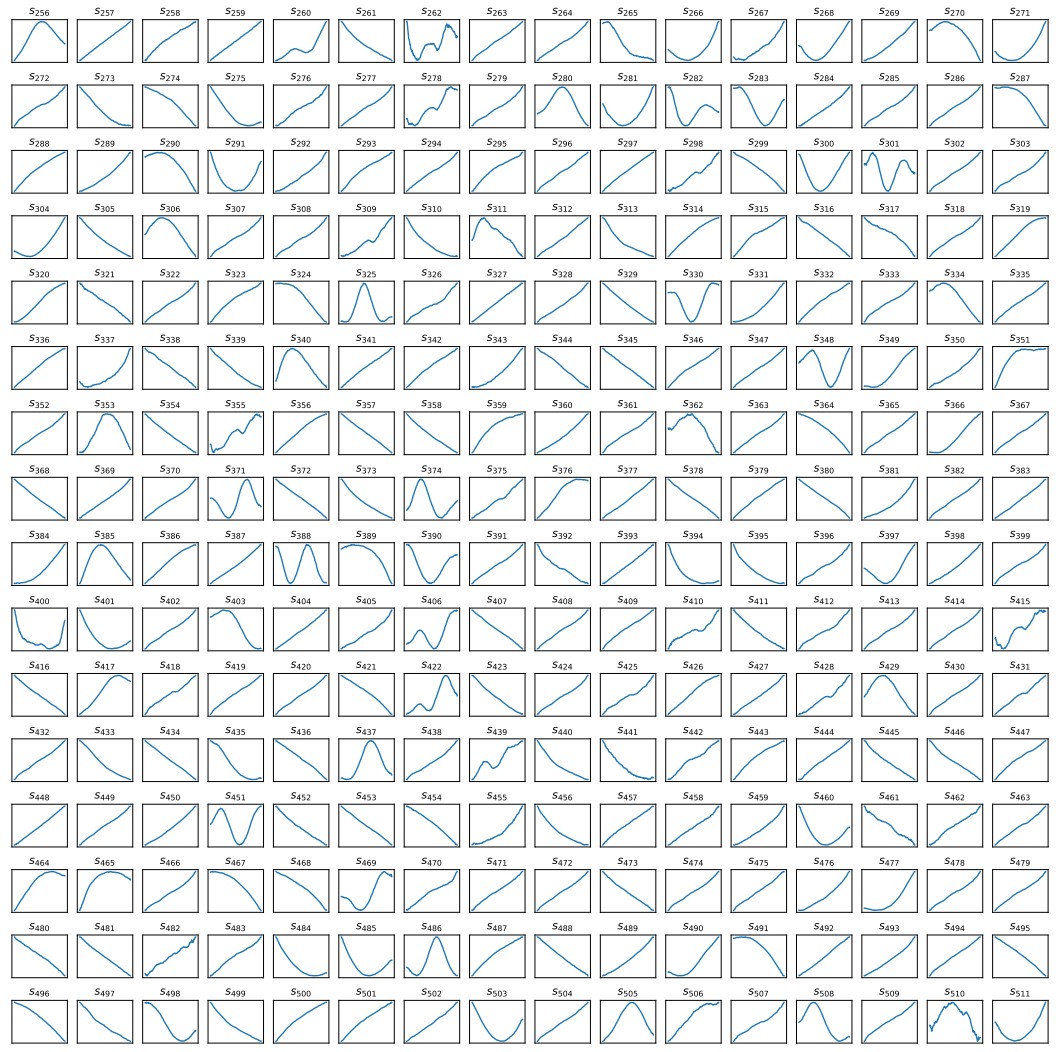

Figure 6: (Continue) Visualizations of the abstracted shapes decoded from the codebook of VQShape

## C.2 Code Distribution

In Figure 7, we plot the distribution of 512 codes in the codebook transformed to 2D space using t-SNE [van der Maaten and Hinton, 2008]. The codes can be roughly cluster into 60 groups, suggesting that there may only exists 60 diversed abstracted shapes in the codebook.

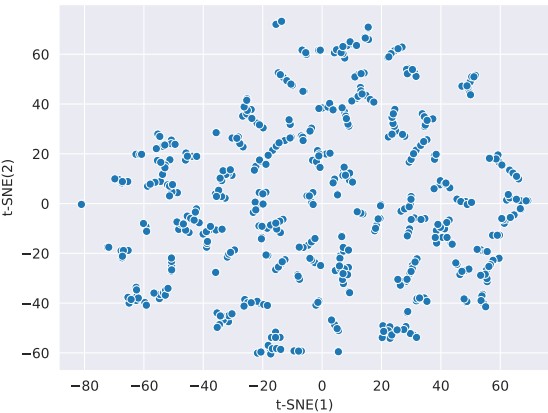

Figure 7: t-SNE plot of the codes.

## C.3 Visualization of $\kappa$ transform

We provide a visualization $\kappa(t, l)$ transform discussed in Section 4.1 in Figure 8. The left figure shows $(t, l)$ samples uniformly sampled from the original coordinate. According to Section 3.1, $(t, l)$ samples can only appear in the lower-triangular plane. After the $\kappa(t, l)$ transform, the corresponding samples are plotted in the transformed coordinate. Samples with small $l$ in the original coordinate becomes more separated in the new coordinate, which encourage the model to capture local details in short subsequences. Samples with large $l$ becomes more concentrated and are less sensitive to their $t$ value since they are likely to capture redundant information.

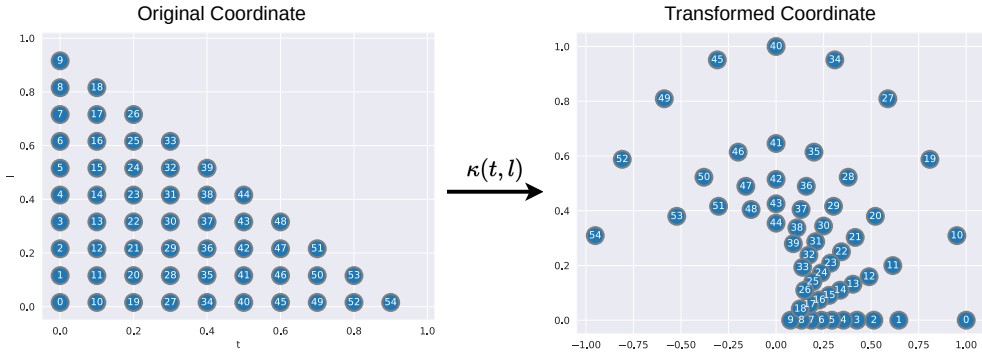

Figure 8: Visualization of transformation $\kappa(t, l)$.

