# OpenReview forum: "Abstracted Shapes as Tokens - A Generalizable and Interpretable Model for Time-series Classification"
_NeurIPS.cc/2024/Conference — NeurIPS 2024 poster_

### Official Review · Reviewer_mtPy · 2024-07-04

**Soundness:** 3
**Presentation:** 2
**Contribution:** 2
**Rating:** 4
**Confidence:** 4

**Summary:**

This work aims to provide an interpretable and generalizable model, called VQshape, for learning on time-series. The key idea of the work is to learn representation based on shapelets, or shape-level features of timeseries as quantized vectors. By learning such a codebook, VQshape can learn in a dataset-agnostic and interpretable way. The authors pre-trained the model on diverse datasets and showed that the proposed model achieves comparable performance as other models on UEA classification benchmark. They also studied the interpretablity and generalizability of the proposed method.

**Strengths:**

- Interesting idea. Using the concept of shapelet to learn time series may provide more interpretable models.

**Weaknesses:**

1. While the high-level idea is interesting, I am not convinced that the model worked.
- Method 3.1 & 3.2: You use a transformer model to produce attribute tuples, why and how could you guarantee the produced attributes are accurate? PatchTST type of architectures fundamentally have this architetcure-level bias that breaks the shape of timeseries during the tokenization stage, how did you mitigate this side-effect and learn real shapes inside data?
- In equation 5, is your subsequence a latent subsequence? If the answer is yes, how did you ensure the target latent subsequence has sufficient fidelity? If the answer is no, isn't your final reconstruction loss just a multiplication of the first term?
- Line 188, what is the point of using shapelets if your model needs to interpolate all the input univariate? Did you consider that even in the same dataset, the length of the input may vary, and naively interpolating them would further mess up the sampling rate, even within the same dataset?
- Figure 4 -- The learned codes are highly identical to each other. There lack high-frequency components in your constructed codebook.

2. Experimental results are not convincing.
- The proposed method has a lot of parameters, a lot of pre-trainings, yet it only performs similarly as TimesNet (which is much smaller and does not need pretraining).
- Lack of other models as baselines.
- Lack of experimental results as a whole. Can the proposed model be used for other tasks, e.g. regression, imputation, anomaly detection?

3. The writing has serious clarity issues. Two examples below:
- grammar error in Line 17
- "token" in transformer has a very specific meaning. Yet the paper use the same word token for quantized vectors from the codebook.

4. Lack of related works about interpretability.
- There exist other ways to build interpretability into time series models. Common methods include: (1) Visualization of attention map across multi-scale transformer features [1]; (2) Use added tokens to "prompt" transformers for specific purpose [2]. When benchmarking the interpretability of the proposed approach, the authors should consider these methods.
[1] Dhariwal, P., Jun, H., Payne, C., Kim, J. W., Radford, A., & Sutskever, I. (2020). Jukebox: A generative model for music. arXiv preprint arXiv:2005.00341.
[2] Xiao, J., Liu, R., & Dyer, E. L. GAFormer: Enhancing Timeseries Transformers Through Group-Aware Embeddings. In The Twelfth International Conference on Learning Representations.

**Questions:**

NA

**Limitations:**

Yes, they are mentioned in section 6

---

> ### Author Rebuttal · Authors · 2024-08-07
>
> Thank you very much for taking the time to carefully read our paper. Below, we address your questions.
>
> > Regarding "why and how could you guarantee the produced attributes are accurate?"
>
> It is important to note that our proposed method is a self-supervised pre-training model, and there are no labeled subsequences to fit. Therefore, "accurate attributes" is not a valid metric here. We can validate that the model produces "good" attributes since the decoded shapes are close to the time-series for samples not included in pre-training (see Figure 2 for an example).
>
> > Regarding "architecture-level bias of PatchTST model"
>
> PatchTST breaks time-series into patches and projects the patches into embeddings. However, as it is a Transformer with full attention, each embedding contains non-local information. Therefore, the authors do not think that this can be a side effect that affects our proposed idea and method. The subsequence reconstruction loss in Equation 5 minimizes the difference between predicted shapes and real subsequences, which encourages learning real shapes.
>
> > Regarding "Equation 5 and loss function"
>
> A latent code is decoded to a subsequence in the time domain. Equation 5 penalizes the difference between decoded shapes and real subsequences in the time domain. The authors are confused by the comment "reconstruction loss just a multiplication of the first term." Equation 4 penalizes the difference between the predicted time-series reconstructed from a set of $\tau_i$ and the input time-series, and Equation 5 penalizes the difference between a single predicted shape and a real subsequence. Additionally, they train different components in the model, where Equation 4 trains the time-series decoder $\mathcal{D}$ and Equation 5 trains the shape decoder $\mathcal{S}$; and gradients from them back-propagate to train the codebook and encoder.
>
> > Regarding "interpolating the input time-series"
>
> We argue this comments with three points:
> - Related works: It is common in time-series models to interpolate the time-series. For example, MOMENT subsamples long sequences to 512 timestamps and pads the short ones, and TimesNet unifies the lengths of input series to perform convolutions. Most of the common baseline methods (summarized by Wu et al., ICLR 2023) do not explicitly consider the length and sampling rate of data. Therefore, we do not think interpolating the data will create a fundamental defect in time-series classification tasks.
> - For our method: Building generalizable models requires determining a generalizable way to model time-series data, such as predicting the missing patch in MOMENT and predicting the future in TimeGPT-1. In our case, we view the method of describing a time-series with 64 abstracted shapes as the generalizable way to model time-series data. Since the abstracted shapes and their attributes are defined in relative scales, we can interpolate the time-series into the same length to improve execution efficiency.
> - Difference with "shapelets": In this paper, we term our codes and decoded sequences as "shapes" instead of "shapelets" since "shapelets" are originally defined as exact subsequences from the original time-series data, while our "shapes" only represent certain trend patterns. The abstracted shapes are agnostics to sampling rate and sample length. We don't think using them after interpolating the data could be an issue.
>
> > Regarding "redundant shapes in codebook and high-frequency codes"
>
> Our current model is trained with a codebook size of 512. The codes in Figure 4 are learned from pre-training data. From the hindsight result, we see that many codes are decoded into similar shapes, which indicates that the codebook size can be reduced. We provide additional results on the model trained with smaller codebook sizes (see the Rebuttal attachment). Learning codes with high-frequency components is not guaranteed as classification datasets either do not contain high-frequency data or the high-frequency components cannot provide shape-level features. Additionally, modeling high-frequency components of time-series can be addressed by multiple tokens. However, we don't think this should be termed as a defect of the model as the codes are purely learned from data.
>
> > Regarding "comparison with TimesNet"
>
> TimesNet is a dataset-specific model trained with supervised learning. Our model is pre-trained on multiple datasets and the weights are frozen when adapted to specific datasets with linear probing, where the linear classifier can have much fewer parameters than TimesNet. Providing useful representations that can efficiently adapt to downstream tasks is the purpose of using pre-trained models such as VQShape and MOMENT. Additionally, our method provides interpretable and generalizable representations, while TimesNet is a black-box model with end-to-end predictions.
>
> > Regarding "additional baselines and experiment results"
>
> In the submission, we chose the best methods among each category of methods and compared them with our method. We additionally provide extensive baselines and comparisons in the Rebuttal attachment (refer to global response and Table 1). We focused on classification tasks in this paper since shape-level features can be more significant and informative in classification tasks than in other time-series tasks. We additionally provide preliminary results on extending our method to forecasting and imputation (refer to global response and Table 2 in the Rebuttal attachment).
>
> > Regarding "`the use of term token"
>
> It is not clear what "very specific meaning" refers to, and whether it is specific to NLP. As a reference, in computer vision, VQGAN also uses token to term their codes. The authors of Large Vision Model [Bai et al., 2023] explicitly term the VQGAN encoder as image tokenizer. Therefore, we believe using the term token in our case is appropriate.

---

> ### Author Response · Authors · 2024-08-12
>
> As we approach the end of discussion period, we want to make sure all your concerns are properly addressed. Please feel free to reach out if any additional clarifications are needed to assist you in future discussions and evaluations. Thanks again for your valuable time and feedback.

---

> ### Comment · Area_Chair_UFGA · 2024-08-12
>
> I wish to second this request by the authors. Dear reviewer mtPy, could you kindly comment to which degree the authors' response addressed your concerns and, if not, clarify your remaining concerns further? This would be particularly crucial since there seems to be some disagreement between reviewers on this particular paper.

---

> ### Comment · Reviewer_mtPy · 2024-08-12
> **Thanks**
>
> Thanks for your response.
>
> 1. ""accurate attributes" is not a valid metric here." The paper claims that the attribute tuple is constructed as (code, offset, scale, relative starting position,  relative length). Surely you won't have a ground truth for the code, but why don't you have any ground truth for the offset, scale, relative starting position, and relative length? I am quite confused about the choice to use MLPs to learn the later 4 attributes as well.
>
> 2. "each embedding contains non-local information" I am aware that each embedding contains non-local information, but that is not relevant to the correctness of learning shapelets. Transformers are short-sighted architectures that ignore local information and focus on global information that naturally breaks the shapelets.
>
> 3. "Equation 5 and loss function" -- Got it.
>
> 4. "interpolating the input time-series"
> - It is not common in time-series models to interpolate the time-series **across datasets**. e.g. TimesNet unifies the lengths of input series in each UEA subset to perform convolutions, yet they never interpolate all UEA subsets (from 10+ length to 3000+ length) to the same 512 length.
> - "improve execution efficiency" is an independent factor w.r.t. "learning good shapelets". If the goal is to improve efficiency, it should be ablated.
>
> 5. "redundant shapes in codebook and high-frequency codes". If the authors are arguing that classification tasks in time series in general do not require high-frequency information, that is not true and that shows you are considering a limited subset of tasks. If the authors are arguing that "not learning enough high-freq information is not a defect of the model", I do not agree. If the authors are arguing that "The current data does not have high-freq data, so they are not learnt", I would need to see experimental results that prove this point.
>
> The other responses address my concern with limited experimental results to some extent. However, my major concern is not adequately addressed. Specifically, the paper claims to learn shapelets to improve interoperability, yet (1) the proposed method is built based on aggressive data manipulation during pre-processing, (2) the chosen architecture is quite specific. Based on many works in vision, it is reasonable to believe the chosen architecture might not be a good choice to learn shapelets; (3) According to the visualizations provided in paper, the learnt shapelets have a limited amount of diversity and thus I am not convinced by the interpretability arguments. Based on all above factors, I would keep my score the same given my current understanding.

---

> > ### Comment · Reviewer_mtPy · 2024-08-12
> > **Thanks**
> >
> > Actually, I'd raise my score from 3 to 4. While my concerns remain, and I think the authors should either address them, or carefully discuss them in the revised paper, a score of 3 was a little harsh because the idea itself is still interesting, may encourage interesting follow-ups, and in my batch of paper the other 3s are much worse.

---

> > > ### Author Response · Authors · 2024-08-13
> > >
> > > We appreciate the reviewer’s clarification on the questions raised. Below, we provide further explanations regarding our Rebuttal and submission.
> > > > Regarding "accurate attributes"
> > >
> > > In the self-supervised pre-training process, there is no ground truth for labeled subsequences, meaning that there is no inherent ground truth for shape, offset, scale, relative starting position, and relative length. Instead, the model predicts these attributes, which are then quantized into $(z,\mu,\sigma,t,l)$ as described in Equation 1. As discussed on Line 141, we first identify the subsequence specified by $(t,l)$ and use it as a "pseudo ground truth" to train $(z, \mu, \sigma)$ using Equation 5, while $(t,l)$ are encouraged to capture disentangled shapes through the regularization in Equation 7. Therefore, since these attributes are derived entirely from the model’s predictions, we believe that their accuracy cannot be appropriately measured in this context. However, the value of the subsequence reconstruction loss could be considered as a measure of accuracy for $(z, \mu, \sigma)$. During pre-training, this loss is minimized to the range of 0.25 to 0.3, as we aim to learn abstracted shapes rather than exact matches like shapelets. Overall, we do not think it is feasible to evaluate this self-supervised pre-training process based on "learned attributes are guaranteed to be accurate."
> > >
> > > Regarding the use of MLP: In Equation 1, the attributes are decoded from the embedding $h$ by functions $f$. As is standard practice in many machine learning methods, a shallow MLP is an appropriate choice to implement a simple non-linear function.
> > >
> > > > Regarding "the use of PatchTST backbone"
> > >
> > > It is important to note that the outputs of the Transformer are not directly the shapelets, but their attributes, where the shape decoder $\mathcal{S}$ maps them to abstracted shapes in the time domain (see Line 140 and Equation 3). The patched Transformer serves as a time-series feature extractor, and we regulate its output embeddings to train a mapping between its outputs and abstracted shapes in the time domain (using Equations 1 and 3). Since the model is presented with the full information of the input time-series, and each embedding contains non-local information, we believe the Transformer backbone is a suitable choice for our purpose of producing attributes that can be mapped to shapes in the time domain. Additionally, while we focused on the Transformer backbone in this paper, other backbones (such as ResNet) can easily be adapted as feature extractors for our method. The choice of backbone is not necessarily specific.
> > >
> > > We hope these clarifications address your concerns about the choice of backbone model.
> > >
> > > > Regarding "interpolating the input time-series"
> > >
> > > We agree with the reviewer that interpolating time-series across datasets can result in the loss of information such as original length and sampling rate. However, as a dataset-agnostic model, VQShape learns to describes any time-series using a set of abstracted shapes with their offset, scale, **relative** position, and **relative** lengths, which are unaffected by the information loss during interpolation. Furthermore, the pre-trained VQShape only provides representations, while the downstream classifiers are dataset-specific, where each dataset consists of time-series with the same length for the benchmarks considered in this paper (irregularly sampled time-series is not the focus of this paper). TimesNet does not unify lengths across datasets since its training is dataset-specific, where processing inputs with various lengths is not explicitly considered.
> > >
> > > We hope the clarifications above explain why interpolations do not affect how VQShape describes time-series data, as well as the down-stream dataset-specific tasks. Then, by unifying the lengths, we can process the inputs in batch and make pre-training more efficient.
> > >
> > > > Regarding "high-frequency codes"
> > >
> > > We believe the reviewer may have overlooked our arguments that "high-frequency components cannot provide shape-level features" and that "modeling high-frequency components of time-series can be addressed by multiple tokens," as stated in the Rebuttal. These points are crucial in explaining why high-frequency codes are not learned. For example, code 388 in Figure 4 represents a shape with more than one period, where a high-frequency sequence can be represented by multiple codes with small $l$. Based on this, we argue that "not learning high-frequency codes should not be termed as a defect of the model" because "the codes are purely learned from pre-training data." However, we agree that explicitly modeling high-frequency components could further enhance the model, addressing the limitation we discuss on Line 261, which we will explore in future work.
> > >
> > > We hope our clarifications address your concerns.

---

> > > > ### Comment · Area_Chair_UFGA · 2024-08-13
> > > >
> > > > I thank the reviewer and the authors for taking the time and effort to clarify the points further. This will be very helpful in the decision-making process.

---

> > > > > ### Author Response · Authors · 2024-08-13
> > > > > **Thank you for your efforts**
> > > > >
> > > > > We would like to thank the AC for your time and great effort in arranging the discussions! We hope that we have addressed some of the major concerns raised by the reviewers.
> > > > >
> > > > > **Dear Reviewer mtPy,**
> > > > >
> > > > > Thank you for your feedback on our paper. We’ve tried to address all your concerns in our responses and hope you find them satisfactory. If there’s anything else you’d like us to clarify, please let us know before the end of the discussion period. We value your input and would welcome any additional comments. If our clarifications address your major concerns, we would appreciate your consideration in raising the score. Your support would be important for our work.
> > > > >
> > > > > Best Regards,
> > > > > Authors

---

### Official Review · Reviewer_8MWE · 2024-07-08

**Soundness:** 3
**Presentation:** 3
**Contribution:** 3
**Rating:** 7
**Confidence:** 4

**Summary:**

The paper introduces VQShape, an interpretable pretaining method for time series (TS) classification. VQShape uses transformer-based TS encoding combined with a VQ-VAE style codebook representation. The latter enables the representation of a TS as a set of shapelets. The VQShape encodes shapelets in a generalized form with additional information on length, position, and offset to overcome the typical limitations of shapelets, such as dataset specificity. The experimental evaluation highlights that VQShape performs on par with other pretraining methods while being interpretable and smaller.

**Strengths:**

- The paper introduces the novel method VQShape, which brings interpretability to time series pretraining.
- To do this, VQShape introduces a modular and general way of utilizing shapelets over multiple datasets.
- The paper is well-motivated and follows a clear structure. The writing is understandable and relatively easy to follow.

**Weaknesses:**

- The mathematical notation is not always precise and could need some clarification (specific details in the questions below)
- The figures overall, but specifically Fig. 1, need more extensive captions to improve their comprehensibility.
- The results in Tab. 1 do not include standard deviation information, which makes it difficult to judge the significance of the results and compare the different methods.

**Questions:**

Questions and improvements to the mathematical notation:

- Introducing $l_\text{min}$ in 3.1 would make the representation easier to understand.
- Similarly, providing formulas for $t_k$ and $l_k$ would be beneficial right there.
- Further, for $l_\text{min}$, $t_k$ and $l_k$ It is unclear if these should be integers or real numbers. While the text (and the later usage of them) points to them being relative, i.e., in \[0, 1\], the initial definition in lines 113 and 114, in combination with $T$ does not seem fitting to that. Also, when used to specify a time-series subsequence (e.g., line 142), they should be timestamps instead of being relative. The mathematical notation in this regard should be checked and aligned.
- Relating $\tau$ to $\hat\tau$ near l. 129 would make the section more digestible.
- Overall, more intuition for the mathematics could make the paper even easier to understand.

General remarks and questions:

- For me, the claim that VQShape outperforms MOMENT based on the results in Tab. 1 does not seem justified without further information. While the mean accuracy and mean rank of VQShape are a bit better than those of MOMENT, the differences are quite small compared to the large variations between the datasets overall. Claiming that VQShape outperforms moment would require some significance test, similar to comparing the mean rank to the other methods. (See \[1\] for more details on aggregation of results; see \[2\] for significance testing).
- Regarding the results in Tab. 2, which show that VQShape can perform comparably even when trained on substantially fewer datasets: Do the authors have an intuition of how the other pretraining methods would do when pretrained on just a subset of the datasets?
- Overall, all figures could benefit from better captions to give context to the figure. Each figure + caption should ideally be understandable on its own, or at least to a reasonable degree. In this paper, the figure captions do not provide the necessary context to the reader. Further, In the caption of Fig. 3: the "presnetation" should be "presentation"
- In Figure 3: What are the different channels? Are these variates from the dataset? Further, the top and bottom parts each represent one sample, or did I misunderstand the histograms?
- In the appendix, line 391 states that the standard deviations for Tab. 1 should be somewhere, but the reference is to the wrong table (presumably), and the standard deviations are nowhere in the appendix.
- The clarity and the structure of the references could be improved. They are rather inconsistent (e.g., ICLR is noted differently among several entries) and contain a lot of unnecessary parts (e.g., URLs for published papers)
- The visualization of the entire codebook in Appendix B.2 and B.3 is super insightful. However, many shapelets seem very similar, and their count should be easily reducible, as hinted in lines 264-268. Is it only visible in hindsight and necessary to learn a large codebook first, or could one start out with a smaller codebook right away?
- The method introduces a possibly helpful indictive bias that would be difficult to enforce in deep auto-encoder architectures: It assumes that a TS can be summarized entirely by shapelets with position, scale, etc. This very implicit influence on the encoder could be discussed better.

References:
- \[1\] Fleming, Philip J., and John J. Wallace. "How not to lie with statistics: the correct way to summarize benchmark results." _Communications of the ACM_ 29.3 (1986): 218-221.
- \[2\] Demšar, J. Statistical Comparisons of Classifiers over Multiple Data Sets. Journal of Machine Learning Research (JMLR), 2006.

**Limitations:**

While the main limitations are discussed in the paper, a short discussion of the trade-off between the codebook size could be beneficial: A larger size means potentially better learning, while a smaller codebook would be better for interpretability.

The main paper or the appendix should briefly discuss the societal impact.

---

> ### Author Rebuttal · Authors · 2024-08-07
>
> Thank you very much for taking the time to carefully read our paper and writing a detailed review. We appreciate your comments and suggestion on giving more comprehensive statistical comparisons with the baseline. Below, we address your main concerns and your questions.
>
> > Regarding "standard deviation of experiment results"
>
> We report the standard deviation of classification accuracy across 5 runs in Table 4 of Appendix B.1. The authors apologize for the incorrect reference in Line 393 that points to Table 2 (a summary of Table 4) for these results. We will fix this in the revised version.
>
> > Regarding "VQShape outperforms MOMENT without further information"
>
> We follow widely used metrics to compare our methods with the baseline methods, including average accuracy, average rank, and number of top-1. Many recent deep learning methods are compared solely based on average accuracy (refer to Wu et al., ICLR 2023, Zhou et al., NeurIPS 2023). The authors agree that the statistical comparisons pointed out by the reviewer are reasonable metrics. We provide additional statistics for comparison in Table 1 of the Rebuttal attachment (also refer to the Global response for details). Based on the results, our method still achieves comparable performance as the SOTA methods in time-series classification and outperform other methods in some metrics, while additionally provides the benefit of producing interpretable features.
>
> > Regarding "performance of other models pre-trained on fewer datasets"
>
> As the pre-training and implementation of MOMENT are not fully open-sourced, we cannot complete this experiment within the tight rebuttal window. We will try to include it in the revised version if accepted.
>
> > Regarding "clarification on Figure 3"
>
> Channels are variates of multivariate time-series, and we will change the text in the figure accordingly. The top row consists of the histograms of the three variates averaged over all samples from the CW category, and the bottom row consists of the histograms of the three variates averaged over all samples from the CCW category. As a clarification, we will improve Figure 3 by replacing "Channel" with "Variate" and revising the caption by adding "Histograms are averaged over all the test samples from the CW and CCW category, respectively."
>
> > Regarding "codebook size"
>
> We previously experimented with a heuristically chosen codebook size of 512. The number of "distinct" shapes could depend on the volume and distribution of pre-training data. The model can be pre-trained with a large codebook (such as 512) to guarantee expressiveness for fitting large datasets. However, the actual number of "distinct" shapes is a statistical pattern learned from pre-training, where such a conclusion can only be made afterward. We studied the effect of reducing the codebook size on downstream classification tasks and summarized the results in the Rebuttal attachment (see Table 2 and the Figures). Please refer to the global response for details. These results indicate that one can start with a small codebook size but may sacrifice performance if the chosen size is too small to fit the data.
>
> > Regarding "inductive bias that would be difficult to enforce in deep auto-encoder architectures"
>
> Based on our understanding of this comment, we conducted an ablation by removing the subsequence reconstruction loss and reported it in Table 2 of the Rebuttal attachment (see the column with $\lambda_s=0$). With representations produced by the model pre-trained with $\lambda_s = 0$, the classifiers result in lower accuracy, indicating that the shape reconstructions actually provide additional useful information. We believe this serves as strong evidence of our advantage. However, as the shapes are abstracted subsequences, we observe that $\mathcal{L}_s$ cannot be effectively minimized to a very small value during pre-training (stuck around 0.3), which could reflect the reviewer's point on "difficult to enforce".
>
> We thank the reviewer again for their feedback and hope that we have addressed their questions. If so, we hope they will consider increasing their score. Please let us know if our clarifications require further discussion.

---

> ### Comment · Reviewer_8MWE · 2024-08-09
>
> We thank the authors for the response and appreciate the clarifications.
> > Comparisons to other methods.
>
> We appreciate the other results for the other baselines. What exactly does the p-value in Tab.1 refer to? Specifically, with regard to what metric is it computed?
> Besides that, I understand that other papers make the same mistakes when comparing methods over multiple datasets. However, this does not change the fact that the additional metrics provided only give very limited insights into the actual performance difference between the methods on the datasets.
> As mentioned in the original review, I think it is completely fine not to show that VQShape outperforms the other methods, as it has clear advantages in terms of interpretability. However, the discussion of the experimental results in the paper should reflect this.
>
> > Baseline performance on subsets of the training datasets.
>
> I agree that the rebuttal phase is not the time to perform such an experiment. This is why I asked for an intuition about the baseline performance in my original question, but I think this got lost in the answer. Can you provide such an intuition?
>
> > Inductive bias of VQShape.
>
> I think the question got misunderstood. I was not asking for additional experiments but rather a discussion of the point in the paper. In my question, I mentioned that VQShape induces an inductive bias, i.e., it is possible to represent the time series via multiple shapelets. Your method induces this bias in a deep autoencoder, which evidently helps to train an interpretable model. However, the fact that VQShape induces this bias is not really discussed in the paper. In that sense, the additional ablation does not really answer the question/comment, but rather a (brief) discussion of this point should be added to the paper.
>
> > Remaining questions and comments.
>
> While your response answered several of the initial questions, some points of my initial review were left out. This includes, for example, comments and questions regarding math notation, figure captions, etc. Could the authors briefly comment on these (e.g. clarify math notation questions ...)?

---

> ### Author Response · Authors · 2024-08-09
>
> Thank you for the follow-up comments and clarifications. Below, we provide additional clarifications on our rebuttal.
> > Regarding "Comparisons to other methods"
>
>  The p-value is obtained from the Wilcoxon signed-rank test, which compares the rank of baseline methods with VQShape (the last column) across the UEA datasets. The p-value ranges from 0 to 1, with a small p-value indicating that the two methods (a baseline and VQShape) are significantly different, and a large p-value suggesting that the methods perform similarly on the datasets. We agree with the reviewer’s comment that the statistical significance is not strong enough to claim superior performance. Therefore, we will revise our claims in the paper to state, "VQShape achieves comparable performance with SOTA baselines while additionally providing the benefit of producing interpretable features."
>
> > Regarding "Baseline performance on subsets of the training datasets."
>
> Thank you for the clarification. We will include more analysis on this question in the revised version, as well as experiment results to support them if possible. Our insights on this question can be summarized as:
> - The pre-training datasets play an essential role in determining the quality of representations from pre-trained models, where such observations have been made in pre-trained NLP and Computer Vision models. Therefore, if pre-trained on the same datasets and having the same backbone (e.g., MOMENT and VQShape both take patch-based Transformer as the backbone), we expect the models to have similar performance on down-stream tasks.
> - The difference in pre-training objective could be an important factor. MOMENT and TST employ the masked-autoencoding objective in pre-training (similar to BERT and Vision-Transformer), where each embedding will tend to capture local information. In VQShape, as we use low-dimensional code and introduce the subsequence reconstruction loss (Equation 5), the tokens and representations will learn more structured, concentrated, and non-local information. Additionally, the ablation experiment on setting $\lambda_s = 0$ is an evidence that demonstrates the subsequence reconstruction loss introduces beneficial information. Therefore, we think VQShape may also slightly outperform other pre-trained models in generalization if pre-trained on subsets of pre-training datasets.
>
> > Regarding "Inductive bias of VQShape."
>
> We thank the reviewer for this insightful comment and apologize for any misunderstanding in our previous response. In the revised version, we will address this question by discussing the following points:
> - The encoder of VQShape introduces an inductive bias that represents and summarizes univariate time-series data using a set of abstracted shapes along with their position, length, offset, and scale.
> - The pre-training objectives guide the encoder toward learning interpretable (subsequence reconstruction in Equation 5) and disentangled (regularization in Equation (7) representations, while preserving the information necessary to describe the time series (reconstruction in Equation 4). These objectives mitigate the typical limitation of deep autoencoders, which often lack interpretability.
> - By pre-training on diverse datasets with a universal codebook, VQShape further leverages the inductive bias to be discrete and dataset-agnostic.
>
> We hope that these additional discussions will address your concerns and enhance the overall quality of our work.
>
> > Regarding "clarification on math notations"
>
> We thank the reviewer for these comments to improve the clarity of our presentation and notations. In the revised version, we will clarify them as follow:
> - Move line 132 to Section 3.1 and clarify "We set $l_{\text{min}}=1/64$ as it is the length of a patch."
> - We think keeping the formulas for $t_k, l_k$ in Equation 1 of Section 3.2 is appropriate since the formulas are how the attributes are computed in the model; they are only scalar attributes in definitions in Section 3.1 which do not have a formula.
> - $l_{\text{min}}, t_k, l_k$ are real numbers in relative scale. We apologize for the misalignment in notations. Line 113 will be updated as $0 \leq t \leq 1-l_{\text{min}}$ and Line 114 will be updated as $l_{\text{min}} \leq l \leq 1-t_k$. For simplicity in notations, we will clarify on Line 106 with "In this paper, $x^m_{i, t_1:t_2}$ denotes a subsequence between timestamp $\lfloor T t_1 \rfloor$ and $\lfloor T t_2 \rfloor$ where $t_1, t_2\in [0,1]$ are relative position."
> - On Line 129, we will clarify that $\hat{\tau}$ represents $\tau$ before quantization.
>
> Please let us know if our clarifications require further discussion.

---

> > ### Comment · Reviewer_8MWE · 2024-08-12
> >
> > Thank you for your extended and extensive answers. They clarified the questions I had.
> >
> > With the changes discussed in the rebuttal, this is a substantial step towards interpretable time series pretraining. **I, therefore, raise my score from 5 to 7.**

---

> > > ### Author Response · Authors · 2024-08-12
> > > **Thank you**
> > >
> > > Thank you very much for taking our rebuttal into consideration and updating your review. We appreciate your constructive comments.

---

> > > ### Comment · Area_Chair_UFGA · 2024-08-13
> > >
> > > Thank you very much for considering the authors' responses and updating your score. And thank you all for this fruitful discussion.

---

### Official Review · Reviewer_F3fg · 2024-07-11

**Soundness:** 3
**Presentation:** 3
**Contribution:** 3
**Rating:** 6
**Confidence:** 5

**Summary:**

Authors propose a self-supervised model which can be used for classification. Their method learns abstracted shapes which serves as interpretable tokens and an information bottleneck in their modeling architecture. They compare with state-of-the-art methods on standard classification datasets and show promising performance.

**Strengths:**

1. The paper is well written.
2. The idea is conceptually interesting yet simple, and the results are promising. The authors also compare with some state-of-the-art methods.
3. The paper provides interesting insights such as impact of the quality of pre-training data on downstream classification performance

**Weaknesses:**

I think my biggest concern in the paper is on the benchmarking aspect: (1) the authors use UEA repository, but I would also encourage them to use the UCR time series classification repository. As a strech goal, real world and harder time series classification datasets such as PTB-XL or MIT-BIH can also be used for benchmarking, (2) the authors compare with a very limited number of baselines. The MOMENT paper for example compares with a wide range of deep learning and statistical techniques such as TS2Vec, k-NN, etc. and I believe it is important to have these larger scale comparisons in the paper.

**Questions:**

1. Why did you exclude the InsectWingBeats dataset from your analysis? Why not use all the UCR and UEA datasets for classification?
2. You found that a lot of decoded shapes are similar (increasing or decreasing lines), and found ~60 clusters. What was the impact of reducing the size of the code book on the performance of your model?
3. How is the size of the codebook determined, is it always an afterthought?
4. How can you encourage the decoded shapes to be as diverse as possible?

**Limitations:**

The authors discuss some limitations of their approach in sections 5 and 6.

---

> ### Author Rebuttal · Authors · 2024-08-07
>
> Thank you very much for taking the time to carefully read our paper. We are glad you found our method provides interesting insights. Below, we address your main concerns and your questions.
>
> > Regarding "why excluding the InsectWingBeats dataset"
>
> We discussed the reason at Line 393 in Appendix A. The InsectWingBeats dataset contains very short time-series, such as 1 or 2 timestamps, which do not contain any meaningful shape-level features. Considering the number of samples and channels of this dataset is significantly higher than other datasets (refer to Table 3 in Appendix A), we believe this dataset may pollute the pre-training of our model and therefore excluded it from the experiments.
>
> > Regarding "choice of benchmarking datasets"
>
> We chose to perform quantitative evaluation on the UEA datasets because they are a default choice in most recent works [Zuo et al., AAAI 2023; Wu et al., ICLR 2023; Zhou et al., NeurIPS 2023] on time-series analysis and suggested to benchmark deep learning methods (Deep Time Series Models: A Comprehensive Survey and Benchmark, Zhou et al., 2024). We agree that adding UCR can make the experiments more comprehensive. However, the results for all the 128 UCR datasets are not usually available for the baseline methods (which also means including UCR results is not generally a mandatory requirement); obtaining the results will require additional time and we could not finish them within the Rebuttal period. We will include the results of our methods and baselines in the revised version.
>
> > Regarding "number of baselines to compare"
>
> We chose TimesNet, T-Rep, and MOMENT as our baselines since they are the best-performing methods among supervised, representation learning, and pre-trained models. We agree that it is good to include more baselines, and we have included detailed results and comparisons with 14 baseline methods in Table 1 of the Rebuttal attachment (please also refer to the global response). Compared with more baselines, our method still achieves comparable performance as the SOTA methods in time-series classification and outperform other methods in some metrics, while additionally provides the advantage of producing interpretable features.
>
> > Regarding "the impact of reducing the size of codebook"
>
> We studied the effect of reducing the codebook size on downstream classification tasks and summarized the results in the Rebuttal attachment. Please refer to the global response for details. As a summary, pre-trained with codebook size 64, the Histogram representations of VQShape achieve the best average classification accuracy of 0.715, which outperforms the SOTA baselines. This matches our hindsight discovery in Figure 5 of the paper where there are roughly 60 clusters of codes.
>
> > Regarding "how is the size of the codebook determined"
>
>  We previously experimented with a heuristically chosen codebook size of 512. The number of "distinct" shapes could depend on the volume and distribution of pre-training data. The model can be pre-trained with a large codebook (such as 512) to guarantee expressiveness for fitting large datasets. However, the actual number of "distinct" shapes is a statistical pattern learned from pre-training data, where such a conclusion can only be made afterward.
>
> > Regarding "how can you encourage the decoded shapes to be as diverse as possible?"
>
> In this paper, we use the entropy terms in Equation 6 to encourage the usage of all codes in the codebook, which also promotes the diversity of latent codes. To encourage the decoded shapes in the time domain to be diverse, we can apply additional regularization by adding
> $$
> \mathcal{L}\_{\text{div}} = \frac{1}{|\mathcal{Z}|^2} \sum_{z\_{1} \in \mathcal{Z}}  \sum\_{z\_2 \in \mathcal{Z}, z\_2 \neq z\_1} e^{-|| \mathcal{S}(z_1) - \mathcal{S}(z_2) ||_2}
> $$
>
> to the overall loss function, as introduced by ADSN [Ma et al., AAAI 2020]. However, introducing an additional objective may make pre-training more challenging. We will leave it as future work as it is not the focus of the our current method.
>
> We thank the reviewer again for their feedback and hope that we have addressed their questions. If so, we hope they will consider increasing their score. Please let us know if our clarifications require further discussion.

---

> ### Author Response · Authors · 2024-08-12
>
> As we approach the end of discussion period, we want to make sure all your concerns are properly addressed. Please feel free to reach out if any additional clarifications are needed to assist you in future discussions and evaluations. Thanks again for your valuable time and feedback.

---

> > ### Comment · Area_Chair_UFGA · 2024-08-12
> >
> > I wish to second this request by the authors. Dear reviewer F3fg, could you kindly comment to which degree the authors' response addressed your concerns and, if required, ask for further clarifications?

---

> > > ### Comment · Reviewer_F3fg · 2024-08-12
> > > **Thank you for the rebuttal!**
> > >
> > > I really appreciate the time and effort that the authors have put into the rebuttal. In light of the rebuttal, I raised my score to reflect my current assessment of the paper.
> > >
> > > Here are some suggestions that I believe to improve the paper:
> > > 1. **Choice of benchmarking datasets:** I do not agree with the authors' rationale to use UEA datasets alone. The papers that the authors have cited (TimesNet and GPT4TS) use a subset of 10 datasets from UEA. It is unclear why those 10 particular datasets were chosen. While I appreciate that the authors used all the UEA datasets (instead of a subset), I maintain my recommendation to compare with more classification specific baselines on all UCR and UEA datasets as they are de facto benchmark for time series classification. In fact, many of the baselines that the authors have compared against, namely MOMENT, TS2Vec compare against many time series classification methods on both UCR and UEA datasets. The performance metrics are also publicly available (Table 10 and 11 in [TS2Vec](https://arxiv.org/pdf/2106.10466), and [results](https://github.com/hfawaz/dl-4-tsc/tree/master/results) in the Deep Learning for Time Series Classification Survey [1].
> > > 2. Also in Table 1, the supervised methods that authors have compared against are time series forecasting methods, instead of classification methods. I would recommend that the authors compare some supervised methods from [1] which are tailored for classification.
> > > 3. For future experiments, I found the [pre-training code](https://github.com/moment-timeseries-foundation-model/moment-research) of MOMENT in the public domain.
> > >
> > > However, I really appreciate the effort that the authors have put into the rebuttal, and hope that they would make changes to their manuscript to reflect atleast some of these suggestions.
> > >
> > > ### References
> > > 1. Ismail Fawaz, Hassan, et al. "Deep learning for time series classification: a review." Data mining and knowledge discovery 33.4 (2019): 917-963.

---

> > > > ### Author Response · Authors · 2024-08-12
> > > > **Thanks for your follow-up comments.**
> > > >
> > > > We appreciate the valuable feedback from the reviewer.
> > > >
> > > > > Benchmarking Datasets
> > > >
> > > > The authors fully agree with the reviewer where the UCR datasets should be included for comprehensive benchmarking. We will obtain the results for our models as well as the baseline methods and include them in the revised paper.
> > > >
> > > > > Baseline Method
> > > >
> > > > We acknowledge that the additional baselines are mostly designed for forecasting or general-purpose time-series modeling; but we include them since they are implemented under a [well-established benchmarking framework](https://github.com/thuml/Time-Series-Library) which makes them feasible to reproduce within the Rebuttal period. We will definitely consider including methods designed for classification as baselines.
> > > >
> > > > Thank you very much for taking our rebuttal into consideration and updating your review. We appreciate your constructive comments.

---

> > > > > ### Comment · Area_Chair_UFGA · 2024-08-13
> > > > >
> > > > > Thank you very much for this fruitful discussion!

---

### Official Review · Reviewer_8xHF · 2024-07-14

**Soundness:** 3
**Presentation:** 3
**Contribution:** 3
**Rating:** 6
**Confidence:** 3

**Summary:**

The paper introduces VQShape, a model designed for time-series (TS) data representation learning and classification. VQShape provides interpretable and generalizable representations by utilizing vector quantization to create a codebook of abstracted shapes. These shapes represent low-dimensional codes that describe time-series data across various domains. VQShape represents time-series data by decomposing TS subsequences into attributes like abstracted shape, offset, scale, start time, and duration. This method allows for the creation of interpretable and dataset-agnostic features. The representations derived from VQShape can be employed to construct interpretable classifiers that achieve performance on par with specialized models tailored for specific tasks. VQShape also demonstrates robust generalization capabilities in zero-shot learning scenarios, where it can effectively handle datasets and domains that were not encountered during the pre-training phase.

**Strengths:**

- The paper introduces VQShape, a novel approach to time-series data representation that leverages vector quantization to create interpretable and generalizable representations. The concept of using abstracted shapes as tokens for time-series modeling is innovative, particularly in how these shapes are linked to the latent space features.
- The paper is well-structured and articulates the motivations, methodology, and findings clearly.
- Experimental results show the effectiveness of the approach when compared to baselines and other recent approaches.
- The paper focuses on the task of learning represention for time series classification which is important in many domains.
- The paper performs clear ablations to show the effectiveness of the proposed model components.

**Weaknesses:**

- The paper shows some interesting results on generalization towards unseen datasets but its not immediately clear how it compares to other approaches like MOMENT or TST. I would recommend performing this evaluation for other approaches too.
- The paper can provide more ablations particularly on subsequence reconstruction loss as its one of the important component of the model.
- How does the performance change when dimension of code size is increased?
- Other clarification questions: how to define l_min? how to get h_k from tau_k?

**Questions:**

- Performance comparison on zero-shot evaluation with other approaches.

**Limitations:**

Yes

---

> ### Author Rebuttal · Authors · 2024-08-07
>
> Thank you very much for taking the time to carefully read our paper, recognizing our contributions and giving valuable feedback. Below, we address your questions.
>
> > Regarding "Generalization experiment of MOMENT and TST"
>
> The authors thank the reviewer for this suggestion. However, as the pre-training and implementation of MOMENT are not fully open-sourced, we cannot complete this experiment within the tight rebuttal window. We will try to include this experiment in the revised version if accepted.
>
> > Regarding "Ablation on subsequence reconstruction loss"
>
> If removed, the codes will not have a connection with shapes in the time domain, and the model will be similar to a regular VQVAE, losing the ability to provide interpretable features. We conducted this ablation study and reported the new results in Table 2 of the Rebuttal attachment (see the column with $\lambda_s=0$). With representations produced by the model pre-trained with $\lambda_s = 0$, the classifiers result in lower accuracy, indicating that the shape reconstructions actually provide additional useful information. We believe this serves as evidence of our advantage.
>
> > Regarding "increase code size"
>
> Our motivation for using low-dimensional codes is stated in Lines 135 to 138. Increasing the code dimension will enlarge the bottleneck and possibly create a defect that the codes contain hidden information beyond the decoded shapes. We conducted this experiment and reported it in Table 2 of the Rebuttal attachment (see the column with $d^{\text{code}}=32$). Increasing the code dimension does not significantly improve the performance of classifiers. We acknowledge that the scaling pattern for code dimension may require more extensive study, but we are unable to complete additional experiments within the Rebuttal period.
>
> > Regarding "$l_{\text{min}}, h_k, \tau_k$"
>
> $l_\min$ can be viewed as "the minimum length of a meaningful shape." We chose $l_\min = 1/64$ since the input series are transformed into 64 patches, indicating that a series with a length of $1/64$ is the basic unit for expressing a time-series. As mentioned in Line 148, $h_k = \texttt{Linear}(\tau_k)$, which is a linear embedding layer to convert $\tau_k$ into embedding $h_k$.
>
> > Regarding "Performance comparison on zero-shot evaluation with other approaches"
>
> There are a limited number of pre-trained time-series models applied to classification tasks, as many of them are designed for forecasting (e.g., Google-TimesFM, IBM-TTM, etc.). Due to limitations in published results and open-sourced implementation (MOMENT released a checkpoint but not the pre-training code) we are unable to reproduce the results within the Rebuttal period. We will additionally include a comparison with GPT4TS [Zhou et al., NeurIPS 2023] in the revised version if accepted.
>
> We thank the reviewer again for their feedback and hope that we have addressed their questions. If so, we hope they will consider increasing their score. Please let us know if our clarifications require further discussion.

---

> ### Author Response · Authors · 2024-08-12
>
> As we approach the end of discussion period, we want to make sure all your concerns are properly addressed. Please feel free to reach out if any additional clarifications are needed to assist you in future discussions and evaluations. Thanks again for your valuable time and feedback.

---

### Author Rebuttal · Authors · 2024-08-07

The authors would like to thank the reviewers for taking the time to review our manuscript and for providing constructive feedback. Here, we aim to address and clarify several key points raised by multiple reviewers, and explain added results and figures in the Rebuttal attachment.

> Regarding "comprehensive comparison with more baselines"

In the submission, we mainly benchmark our method against TimesNet, T-Rep, and MOMENT as they are reported by literature to be the SOTA of supervised learning, unsupervised representation learning, and pre-trained models respectively. We did not include more baselines and their results on the UEA datasets are not always available and we did not finishing reproducing them at the time of submission. We include a more extensive comparison with 10 additional baselines using more metrics in Table 1 of the Rebuttal attachment. We can conclude the best performing baselines on the UEA datasets are TS2Vec, T-Rep, TimesNet, Reformer, and MOMENT. While there is no dominant method that consistently outperforms others across all the metrics, our method achieves the best mean accuracy. Beyond comparable performance with the SOTA methods, our method additionally provides the benefit of producing interpretable and dataset-agnostic representations for time-series data.

> Regarding "performance of model trained with different codebook size"

We conduct a more extensive study on the effect of these hyperparameters by evaluating the models pre-trained with different codebook size. We summarize the results in Table 2 of the Rebuttal attachment. Figure 1 in the Rebuttal attachment visualizes the relationship between codebook size and average classification accuracy over the 30 UEA datasets. From these results, we can conclude that using the histogram representation produced by the model trained with codebook size 64 and code dimension 8 results in the best performance on the UEA datasets. However, the performance of using token representations increases as codebook size increases, and the classifiers using histogram representations outperform the token representations when the codebook is small. This indicate that the linear classifier cannot effectively manage the token representations since they are more expressive then histogram representations. Additionally, we visualizes the shapes decoded from codebook in Figure 2 and latent code distribution in Figure 3 of the Rebuttal attachment. We can see that the decoded shapes have lower redundancy compared Figure 4 in the paper.

> Regarding "extension to other time-series tasks"

As an extension to the original submission, we have obtained preliminary results of applying VQShape in both zero-shot and fine-tuning to other time-series tasks including imputation and forecasting. Table 3 and Table 4 in the Rebuttal attachment benchmark the performance of VQShape against MOMENT and TimesNet on imputation tasks and forecasting tasks, respectively. On imputation tasks, VQShape significantly outperforms MOMENT in both zero-shot and fine-tuning settings, and achieves comparable but slightly worse performance than TimesNet. On forecasting tasks, VQShape is able to provide predictions in zero-shot settings but the error can be high. By fine-tuning the model, VQShape achieves comparable performance as MOMENT and TimesNet. These preliminary results demonstrate our statement in Line 302 of the paper where we foresee that VQShape can be extended to general time-series tasks. However, we acknowledge that the results obtained by fine-tuning VQShape is not a fair comparison with results reported by MOMENT using linear-probing, where we aim to conduct more comprehensive and appropriate benchmarking in future works.

---

### Decision · Program_Chairs · 2024-09-25

**Decision:**

Accept (poster)

**Comment:**

The paper proposes a novel vector quantization approach to learn interpretable shapes in time series that not only enhance interpretability but also generalize to other data sets. Reviewers appreciated the novelty of the approach and the clear writing of the paper. Given the complexity of the notation and the experimental evaluations, the reviewers also raised numerous technical concerns and questions. However, these could be addressed during an extensive rebuttal and discussion period, such that multiple reviewers raised their scores. Overall, the consensus after rebuttal appears to be acceptance.